# FedEMoE: Improving Personalization on Heterogeneous Federated Learning via Elastic Mixture of Experts Architecture

## Abstract

Heterogeneous federated learning (HtFL) has emerged as a promising approach to address heterogeneity in local computational resources and data distribution, as is common in the real world. However, existing methods cause performance degradation of model personalization because personalized and generalized knowledge are either intertwined or dominated by one of them. To address this issue, we propose a novel **E**lastic **M**ixture **o**f **E**xperts (EMoE) architecture on HtFL, namely FedEMoE, decoupling personalization from generalization. In detail, FedEMoE employs a multi-scale feature extraction mechanism via personalized experts to enrich personalized knowledge. Furthermore, we design an elastic shared expert to break the transferred knowledge bottleneck across heterogeneous client models. The elastic shared expert can adaptively expand or shrink according to the status of each expert by the weight spectrum analysis, respectively. Moreover, the sparsity of mixture of experts (MoE) alleviates the loss of personalized knowledge that typically results from dense model aggregation. Extensive experiments across statistical and model heterogeneity settings demonstrate that FedEMoE significantly outperforms state-of-the-art federated learning methods on the performance of each heterogeneous model over diverse datasets.

## 1 Introduction

Heterogeneous federated learning (HtFL) has emerged as a promising paradigm for enabling multiple clients to collaboratively enhance their local heterogeneous model where data is naturally distributed and highly heterogeneous (Li et al., 2020; Kairouz et al., 2021). HtFL struggles with statistical heterogeneity, where the server's average aggregation becomes a poor direction for clients, so the local model drifts away from its optimal point and ends up with low accuracy. Moreover, model heterogeneity arises when clients may design their own local model architectures to meet individual requirements, where they can hardly cooperate through models or leverage other clients' knowledge. When the above problems are overcome, HtFL can break down data silos and unlocks significant value through effective collaboration of clients.

Existing HtFL methods address these constraints by three ways. Knowledge distillation-based methods (Lin et al., 2020; Wu et al., 2022) enforce alignment on softened logits, which smooths out personalized decision boundaries. Prototype-based approaches (Tan et al., 2022; Zhang et al., 2024) compress local representations into class-wise vectors, discarding fine-grained local structure. Model-based methods (Niu & Deng, 2022; Chen & Chao, 2022) share a common backbone via naive averaging, forcing local models to compromise their personalization for the sake of global agreement. The above HtFL methods often make dilution of personalized knowledge which extracted from local data, resulting in local model drift.

When personalization and generalization objectives are tied to the same set of parameters, any concession to global consensus directly dilutes local personalization, and vice versa, which creates a zero-sum dilemma: advancing one inevitably sacrifices the other (Chen et al., 2024). Existing HtFL methods fall into personalization decline precisely because they primarily transfer generalized knowledge, forcing local updates closer to the global model—a biased prior that conflicts with

local data (Zhu et al., 2024; Meng et al., 2025). Therefore, the ensuing challenge is to decouple transferable knowledge from the global model to correct local updates from this biased prior.

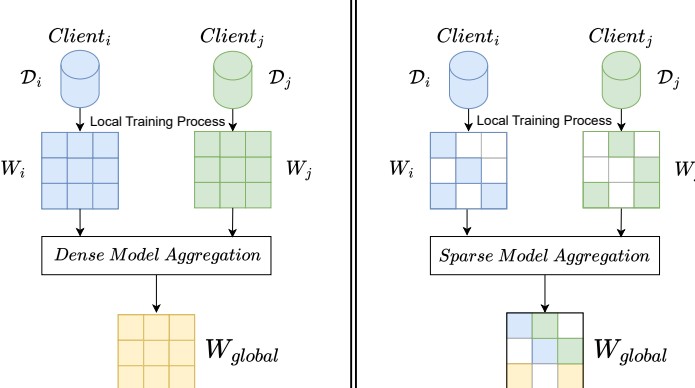

Figure 1: The differences between dense and sparse model aggregation. $D_i$ is the local dataset of client $i$ and $M_i$ is the local model of client $i$.

To escape this zero-sum trap, we first change the structure of the shared model so that it no longer imposes a single, biased prior. The ideal state of an aggregation model is that it is generalized as a whole and personalized in its parts. Therefore, we compare dense with sparse aggregation. As illustrated in Figure 1, the server performs a simple average across a monolithic model in dense model aggregation, which dilutes personalization knowledge. In contrast, sparse model aggregation does not uniformly aggregate models together, which retains whole client-specific weights and thus significantly reduces the loss of personalized knowledge caused by dense model aggregation. This sparse aggregation enables the global model to evolve into a repository of diverse knowledge.

This leads to our core idea: building the system on the personalization objective through decoupling personalization from generalization at the architectural level. Specifically, we redesign the local model with an elastic Mixture-of-Experts (EMoE) architecture, where most experts are not shared modules, but personalized knowledge repositories. The shared experts in EMoE act as a selective knowledge repository, where each client dynamically retrieves and integrates relevant knowledge from others, thereby improving the local model's personalization.

Building on this insight, our **main contributions** are:

- We propose FedEMoE, a HtFL framework for personalization that architecturally decouples personalization from generalization, ensuring local models never compromise their specialization for global consensus.

- We introduce a multi-scale feature extraction and knowledge exchange mechanism in the MoE architecture. This mechanism maximizes the feature captured from local unique data and retained the ability to exchange knowledge with other clients, resulting in the improvement of personalization.

- We design an elastic MoE architecture where the shared expert acts as a dynamic repository of collective intelligence. Its structure can adaptively strengthen its representation capacity by preserving and integrating specialized knowledge rather than averaging it away.

- We evaluate FedEMoE in settings with significant statistical and model heterogeneity. Our extensive experiments and ablation studies demonstrate that FedEMoE is superior to state-of-the-art methods, improving prediction accuracy by up to 48.05%.

## 2 RELATED WORK

We classify the existing methods of HtFL into three categories: *knowledge distillation-based, prototype-based* and *model-based methods*.

**Knowledge Distillation-Based FL Methods.** Knowledge distillation(KD)-based FL methods use model outputs (logits), so they are suitable for model-heterogeneous scenarios. Some methods, such as FeAUX (Sattler et al., 2021), FedDF (Lin et al., 2020), and DS-FL (Itahara et al., 2021), enable models to learn from others by identifying differences in model outputs on a global dataset. To overcome the limitations of public datasets, some approaches have been proposed: FedGen (Zhu et al., 2021) uses a global generator based on clients' outputs to generate data, and FedKD (Wu et al., 2022), FML (Shen et al., 2020) advance KD-based methods in a data-free manner, where they share a small model as knowledge, rather than relying on a global dataset. However, these methods incur additional computational overhead or require a high-quality public dataset, thereby constraining their applicability, and smooth out specific decision boundaries.

**Prototype-Based FL Methods.** Prototype-based FL approaches utilize local data feature representations. For example, FedPAC (Xu et al., 2023), FedGKT (He et al., 2020), and FedProto (Tan et al., 2022) upload local class-specific representations or prototypes to the server to gain global knowledge. FedTGP (Zhang et al., 2024) introduces a regularization approach to increase the spacing between classes, enhancing prototype distinctiveness. FedSA (Zhou et al., 2025) employs semantic anchors to align global prototypes within a unified feature space. However, these methods incur substantial computational overhead, and discard fine-grained local structure. Moreover, in scenarios with numerous clients, the knowledge from global prototypes or representations becomes less distinctive, significantly affecting training efficiency.

**Model-Based FL Methods.** The main concept of model-based FL approaches is to divide client models into two parts: one for global aggregation and sharing, and another one retained locally for model personalization. For instance, methods like FedGC (Niu & Deng, 2022), FedROD (Chen & Chao, 2022), FedRep (Collins et al., 2021), FedBABU (Oh et al., 2022), and FedAlt (Pillutla et al., 2022) split the model into homogeneous feature extractors and heterogeneous classifiers, enabling data mapping to a consensus feature space and preserving local personalization through classifiers. In contrast, methods such as LG-FedAvg (Liang et al., 2020), and Fedclassavg (Jang et al., 2022) use heterogeneous feature extractors and homogeneous classifiers, achieving global knowledge acquisition by sharing classifiers for unified classification standards. However, these methods force local models to compromise their personalization for the sake of global agreement.

## 3 METHODOLOGY

### 3.1 PROBLEM STATEMENT

We consider a central server and $K$ clients. Each client trains heterogeneous models $M_i$ on non-IID data $D_i$ and shares their knowledge with other clients via the server. Meanwhile, the communication overhead between client and server should be minimized. Therefore, the overall collaborative objective is:

$$\min_{\{M_i\}} \frac{1}{K} \sum_{i=1}^{K} \mathcal{L}(M_i, D_i) \quad \text{subject to} \quad \frac{1}{K} \sum_{i=1}^{K} \mathcal{C}(M_i) < c. \tag{1}$$

where $\mathcal{L}(\cdot, \cdot)$ is the loss function, $\mathcal{C}(\cdot)$ is the communication overhead and $c$ is the communication budget. On the client side, the formula for local training is as follows:

$$M_i = M_i - \eta \nabla_{M_i} \mathcal{L}(M_i, \mathcal{D}_i), \tag{2}$$

where $\eta$ is the local learning rate and the $\mathcal{D}_i$ is the training set of $D_i$.

### 3.2 THE OVERVIEW OF FEDEMOE APPROACH

To achieve the above objective, we design FedEMoE which first employs the EMoE architecture to decouple personalization from generalization as shown in Figure 2.

On the client-side, we first introduce a shared lightweight feature extractor (FE) on each client to keep local updates semantically aligned and aggregation directionally consistent. Subsequently, each client model adopts an MoE architecture that contains one personalized expert group (PEG) and one shared expert (SE). In $\text{PEG}_i$, there are $\mathcal{N}_i$ heterogeneous personalization experts. Due to their different structures, they can capture the multi-scale characteristics of data to promote local

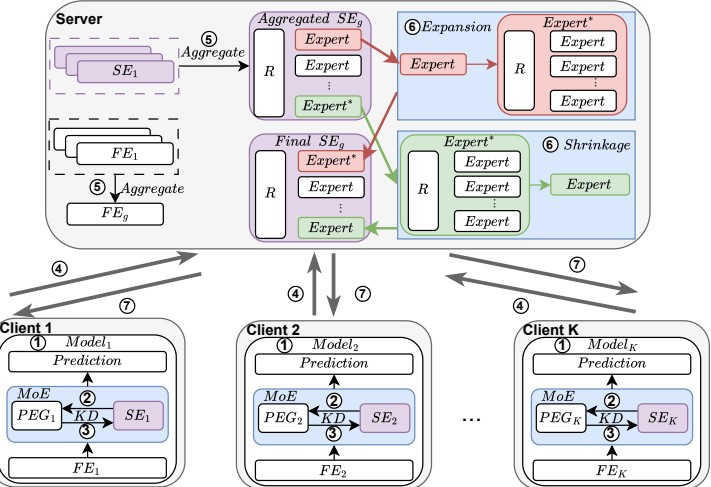

Figure 2: The architecture of FedEMoE. ① Local update on the client-side. ② Personalized expert group (PEG) distills knowledge from the shared expert (SE) on small scale data (only update PEG). ③ SE distills knowledge from PEG on small scale data (only update SE). ④ Participating clients upload feature extractors (FE) and SE. ⑤ The server executes weighted aggregation of FE and sparse aggregation of SE. ⑥ The sub-experts of the aggregated SE respectively execute the expansion or shrinkage according to the result of weighted spectral analysis on the server-side. ⑦ The server sends global FE and global SE to clients.

personalization. Due to the gating network mechanism, different samples activate different personalized expert combinations, resulting in different scale feature combinations for different samples. SE is a dynamic hierarchical structure based on MoE, serving as a carrier for cross client knowledge sharing. Finally, we design a knowledge exchange mechanism to overcome the incomplete knowledge caused by sparse gating in the MoE architecture in Section 3.3.

On the server-side, we aggregate FEs and SEs from participating clients in each round, respectively. Then we propose an elastic MoE architecture to keep the SE always in a "sparse yet high-knowledge-density" state and reduce unnecessary computation and communication costs in Section 3.4. We introduce the knowledge exchange mechanism and the EMoE architecture in detail as follows.

### 3.3 KNOWLEDGE EXCHANGE MECHANISM

Due to the sparse activation characteristic of the MoE architecture on each client, the SE and PEG are unable to all samples from the local data. To solve this difficulty, we introduce a knowledge exchange mechanism between SE and PEG on every client. To prevent gradient oscillation, each side distills the cached output from the other, reducing the process to two independent supervised fittings rather than a minimax game.

**Personalized knowledge fuses to generalized knowledge.** We make the SE to acquire more local personalized knowledge from the PEG on each client. The SE is optimized to match PEG's output distributions for the personalized knowledge fusion. We measure the difference between their output distributions by KL divergence and denote it as $\mathcal{L}_S$. The loss function of this part is defined:

$$\mathcal{L}_S = \sum R_p \log \frac{R_P}{R_S}, \tag{3}$$

where $R_S$ is the SE output, and $R_P$ is the consensus output of all personalized experts in PEG. The consensus output fuses the personalized knowledge into a single target, which guides SE's output towards personalized update direction. The consensus output $R_P$ is calculated as follows:

$$R_P = \sum_{n=1}^{N_i} a_n R_n, \tag{4}$$

where $a_n = \frac{\text{count}(activated\ expert_n)}{\|\mathcal{D}_i\| \cdot topK}$ represents the activation frequency of personalized expert $n$ in PEG, $R_n$ is the output of personalized expert $n$ and $N_i$ is the number of personalized expert in $M_i$.

**Generalized knowledge transfers to personalized knowledge.** In reverse, we optimize a PEG at the objective of the minimal personalized loss $\mathcal{L}_p$ for generalized-to-personalized knowledge transfer. The transfer ensures personalized experts in PEG to learn diverse generalized knowledge from SE. The $\mathcal{L}_p$ is defined as follows:

$$\mathcal{L}_p = \sum_{n=1}^{N_i} \sum R_S \log \frac{R_S}{R_n}. \tag{5}$$

The PEG learns relevant knowledge from the SE for enhancing local personalization on each client.

### 3.4 ELASTIC MoE ARCHITECTURE

Since the PEG learns from the SE, the SE must possess personalized capabilities; however, as a shared-knowledge carrier, it also needs generalization ability. Therefore, we design the SE as a dynamic MoE structure. We perform sparse model aggregation on SE to preserve expert personalization. And we adjust the number of SE's sub-expert to strengthen the SE's knowledge diversity. The elastic MoE architecture works through the following three components.

**Sparse model aggregation of SE.** Once the SE is broadcast, only a subset $\mathcal{K}_j$ of clients that actually activate and update sub-expert $j$. As a result, the next aggregation of sub-expert $j$ is dictated by the consensus gradient of $\mathcal{K}_j$, which enables its representation to stay closer to the data distribution of $\mathcal{K}_j$. After multiple rounds of aggregation, the preferences of each sub-expert tend to stabilize. As a result, sub-expert $j$ can always be activated by $\mathcal{K}_j$'s model and enhance $\mathcal{K}_j$'s local personalization.

**Expert Status Monitoring and Decision Making.** We provide a diagnostic result to guide elastic expansion or shrinkage decisions by monitoring the aggregated status of each expert.

Each client $i$ maintains SE's activation times $B_i = \{b_{i,j}\}$, where $b_{i,j}$ represents the times of local samples that activate sub-expert $j$ in the SE. $B_i$ is aggregated on the server to compute global activation times $B_g = \{b_{g,j}\}$ where $b_{g,j} = \sum_i w_i b_{i,j}$, where $w_i$ is the fraction of client $i$'s local data size to the total data size and $B_g$ serves as a signal for guiding shrinkage on the aggregated SE.

We introduce the weight spectrum analysis technique (Yunis et al., 2025; Martin & Mahoney, 2021) as a diagnostic tool to determine the expert evolution. Unlike validation-loss or gradient-norm triggers, weight-spectrum analysis requires no extra data or tensors and provides a provable drift bound (Appendix D.7.2). Specifically, we perform singular value decomposition (SVD) on the weight matrix $\mathbf{W}^{(l)} \in R^{m \times n}$ of each layer $l$ and sort its singular values in descending order: $\sigma_1^{(l)} \geq \sigma_2^{(l)} \geq \cdots \geq \sigma_r^{(l)} > 0$. We judge the status of experts through two key indicators: the effective rank $p_\theta^{(l)}$ and the tail energy $T_\alpha^{(l)}$. The effective rank $p_\theta^{(l)}$ is defined as the smallest fraction of full rank required to retain an energy ratio $\theta$, where

$$p_\theta^{(l)} = \min \left\{ k \mid \frac{\sum_{i=1}^{k} (\sigma_i^{(l)})^2}{\sum_{i=1}^{r} (\sigma_i^{(l)})^2} \geq \theta \right\}. \tag{6}$$

The tail energy $T_\alpha^{(l)}$ measures the residual energy after discarding the top $\alpha$ proportion of components, where

$$T_\alpha^{(l)} = \frac{\sum_{i=\alpha+1}^{r} (\sigma_i^{(l)})^2}{\sum_{i=1}^{r} (\sigma_i^{(l)})^2}. \tag{7}$$

As shown in Figure 3, we classify experts into three status based on the shape of the spectrum. *Under-fitting* (Figure 3a), *Normal* (Figure 3b), *Over-fitting* (Figure 3c). Specifically, a rapidly decaying spectrum indicates stable capacity with negligible drift, whereas a heavy-tailed spectrum signals redundancy and overfitting.

However, the expert structure may cause redundant computations due to excessive expansion, without any gain in accuracy in some tasks. To address this, we treat the effective rank as a time series $\mathcal{S} = \{S_t \mid t \in T\}$, where $T$ is the total number of communication rounds $S_t = p_\theta^{(l)}(t) \in R$. We

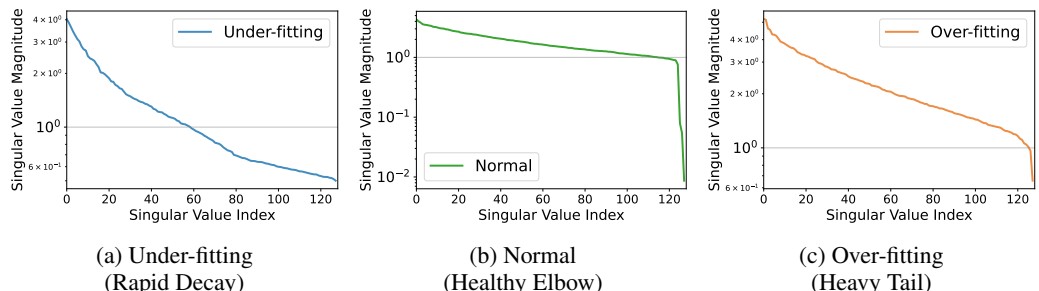

Figure 3: Weight spectrum analysis for expert state classification under CIFAR-100. The singular value decay patterns reveal expert training status: (a) Under-fitting shows rapid decay with low effective rank, (b) Normal exhibits a healthy elbow pattern with balanced capacity, (c) Over-fitting displays heavy tails indicating parameter redundancy. This analysis guides our elastic expansion and contraction decisions.

employ an Exponential Moving Average (EMA) mechanism to dynamically adjust the expansion threshold, thereby adaptively suppressing ineffective expansions. At round $t$, we allow expansion only if $p_\theta^{(l)}(t) > v_t$, thus blocking further growth once the effective rank stops increasing. The recursive formula for the dynamic threshold $v_t$ is as follows:

$$v_t = \begin{cases} v_0, & t = 1 \\ \gamma S_t + (1 - \gamma)v_{t-1}, & t \geq 2 \end{cases} , \tag{8}$$

where $v_0$ is the initial value as a hyperparameter and $\gamma$ is the smoothing factor.

**Experts Evolution.** We design a process of experts evolution to satisfy the richer representation of each SE, where each expert exists in two states: an expert $(e)$ represents a single neural network component with fixed capacity, and a MoE-based expert $(e^*)$ represents a composite structure containing $N$ experts $(e)$ with internal routing.

Expansion $(e \to e^*)$ is triggered when $e$ is diagnosed as under-fitting (low effective rank). The original $e$ becomes $e_j \in e^*$, and the remaining experts are randomly initialized. When the expert $e$ in $e*$ meets the expansion conditions, it can also be expanded. This expansion aims to improve the capacity that the under-fitting expert lacks, preserving existing knowledge while adding expressive power, thus improving local accuracy without extra validation data. The specific expansion process can be found in the Algorithm 3, Appendix C. Discussions on the expansion strategy for initializing new experts can be found in the Appendix D.1.

Shrinkage $(e^* \to e)$ is invoked when the $e^*$ is flagged as over-fitting (heavy-tailed spectrum). The $e^*$ is changed to an expert $e$ via the weighted aggregation. This shrinkage removes redundant parameters, cuts memory and communication costs, while retaining the distilled knowledge of the $e^*$. The specific shrinkage process can be found in the Algorithm 4, Appendix C.

Unlike width scaling(Appendix D.7.1), our approach preserves the existing feature space. The convergence analysis of experts' evolution is presented in Appendix D.5. Through the process, the number of sub-experts can be dynamically changed, and the hierarchical structure of sub-experts enables each sub-expert to serve a small and stable subset of clients.

## 4 EXPERIMENTS

We conduct extensive experiments to validate the effectiveness of the FedEMoE. Specifically, our experiments aim to address the following research questions: (RQ1) How does FedEMoE perform in heterogeneous-model settings? (RQ2) What is its effectiveness in homogeneous-model scenarios? (RQ3) How well does it handle diverse data distributions? (RQ4) How does our approach perform in terms of convergence speed? (RQ5) How scalable is our approach?

### 4.1 SETTINGS

**Dataset.** We test our approach FedEMoE on three datasets: CIFAR10, CIFAR100 (Krizhevsky et al., 2009), and Tiny-Imagenet (Chrabaszcz et al., 2017). These datasets range from simpler to more complex, enabling a structured assessment of our approach.

**Baseline methods.** To evaluate FedEMoE, we focus on the statistical heterogeneity and model heterogeneity settings in federated learning and evaluate it with ten state-of-the-art baselines. Specifically, we include the classic homogeneous-model baseline FedAvg (McMahan et al., 2017); three personalization-oriented homogeneous-model methods, namely the adaptive local-aggregation technique FedALA (Zhang et al., 2023c), the domain-bias-eliminating representation learning approach FedDBE (Zhang et al., 2023a), and the feature-alignment method FedPAC (Xu et al., 2023); the gradient-correction-based personalized algorithm FedGC (Niu & Deng, 2022); and five heterogeneous-model approaches that either distill knowledge (FedKD) (Wu et al., 2022), leverage class prototypes (FedProto,FedSA) (Tan et al., 2022; Zhou et al., 2025), combine meta-learning with representation learning (FedMRL) (Yi et al., 2024), and trainable global prototypes for better generalization (FedTGP) (Zhang et al., 2024).

**Statistical heterogeneity.** In line with prior research (Li et al., 2022; 2021), we introduce statistical heterogeneity among clients via the Dirichlet distribution (Hsu et al., 2019). The process involves sampling $q_{c,i}$ from $\text{Dir}(\beta)$ for each class $c$ and client $i$. Here, $\text{Dir}(\beta)$ denotes the Dirichlet distribution, with the $\beta$ set to 0.1 by default.

**Model heterogeneity.** In our experiments, model heterogeneity mainly consists of the following aspects: differences in model structures caused by the number of personalized experts; the composition of personalized expert groups; the structures of experts and the backbone of models. For detailed model architectures, refer to the Appendix E.2.

**Hyperparameter setting.** In our work, we explore complex scenarios in federated learning. We set the client participation rate of 0.2 per round. This setup allows us to evaluate the performance of various methods when only a small number of clients are active. We adopt one local training epoch on each client per round. We set the batch size at 64 and the learning rate $\eta$ at 0.005, spanning 100 communication rounds ($T$). Each client's local dataset is split into training (75%) and testing (25%) sets. The server performs a scaling operation every $\mathcal{T}$ rounds ($\mathcal{T} = 20$). We use the Adam optimizer for local model updates. For all experiments, we run three trials and present the mean test accuracy. For detailed parameter settings, refer to the Appendix E.1.

### 4.2 IMPACT OF MODEL HETEROGENEITY (RQ1)

We evaluate FedEMoE against the SOTA of HtFL methods. FedEMoE improves accuracy by 32.43% on CIFAR-100 and 40.62% on Tiny-ImageNet. Personalized expert groups learn client-specific multi-scale features, enriching local representations, while elastic shared experts provide a flexible mechanism for rich-knowledge transfer.

Table 1: Test accuracy(%) under different datasets in model-heterogeneous scenarios with K = 50.

| Dataset | CIFAR-10 | CIFAR-100 | Tiny-Imagenet |
|---|---|---|---|
| FedKD | 72.21 | 31.19 | 19.38 |
| FedProto | 68.83 | 25.13 | 2.57 |
| FedMRL | 78.37 | 36.01 | 19.38 |
| FedTGP | 66.47 | 31.96 | 12.21 |
| FedSA | 70.93 | 33.27 | 21.27 |
| **Ours** | **79.83(↑1.9%)** | **47.69(↑32.4%)** | **29.91(↑40.6%)** |

### 4.3 IMPACT OF MODEL HOMOGENEITY (RQ2)

To examine the performance of FedEMoE under model-homogeneous scenarios, we compare it with SOTA baselines. FedEMoE surpasses them by 22.78% on CIFAR-100 and 26.33% on Tiny-ImageNet. A decoupling splits personalized and shared parameters, letting each part learn on its own and removing mutual limits. Elastic shared experts expand the generalized knowledge pool and

broaden universal representations without disturbing any client's personalized knowledge, so local decision boundaries stay intact.

Table 2: Test accuracy(%) under different datasets in model-homogeneous scenarios with K = 50.

| Dataset | CIFAR-10 | CIFAR-100 | Tiny-Imagenet |
|---|---|---|---|
| FedAvg | 76.85 | 35.94 | 23.68 |
| FedGC | 78.68 | 38.71 | 23.73 |
| FedALA | 79.01 | 30.83 | 19.03 |
| FedPAC | 75.21 | 20.82 | 19.81 |
| FedDBE | 75.85 | 36.16 | 20.92 |
| **Ours** | **79.22(↑0.3%)** | **47.53(↑22.8%)** | **29.98(↑26.3%)** |

To demonstrate that FedEMoE accommodates arbitrary architectural differences, we run a backbone-heterogeneous split: clients independently hold ResNet-18, ResNet-34, MobileNet, or GoogleNet, and each inserts its own PEG plus the same narrow SE block. Table 3 reports the mean test accuracy on CIFAR-100 (50 clients). Under this fundamentally heterogeneous setting, FedE-MoE perform better than baselines, proving that our approach does not limit heterogeneity.

Table 3: Double-heterogeneous setting—clients differ both in backbone architecture (ResNet-18/34, MobileNet, GoogleNet) and expert architecture.

| Method | FedKD | FedProto | FedMRL | FedTGP | Ours |
|---|---|---|---|---|---|
| Accuracy (%) | 24.87 | 12.73 | 30.21 | 27.79 | 38.61 |

## 4.4 IMPACT OF STATISTICAL HETEROGENEITY (RQ3)

In order to evaluate the performance of FedEMoE in handling statistical heterogeneity, we conducted a comparative study with other HtFL methods on CIFAR-100 and 50 clients. To control the level of statistical heterogeneity, we utilized the Dirichlet concentration parameter $\beta$, creating scenarios ranging from highly imbalanced data ($\beta = 0.1$) to relatively balanced data distribution ($\beta = 10$). The comparative results are presented in Figure 4. FedEMoE consistently outperforms baselines across all heterogeneity levels, with performance gaps widening under higher imbalance, demonstrating superior robustness to non-IID data distributions.

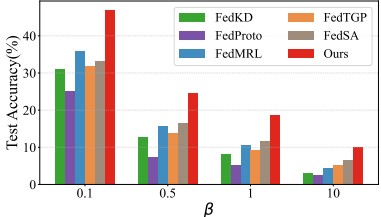
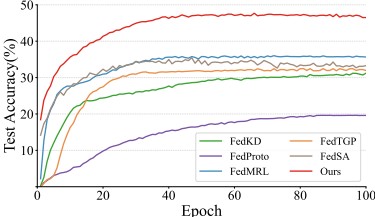

Figure 4: Test accuracy of different methods under diverse statistical heterogeneity.

Figure 5: The test accuracy curves of different methods under heterogeneous settings.

## 4.5 CONVERGENCE SPEED ANALYSIS (RQ4)

To evaluate the convergence speed of our approach, we conducted experiments testing SOTA methods on CIFAR-100 with 50 clients. As shown in Figure 5, FedEMoE significantly outperforms others in both convergence speed and peak test accuracy. FedEMoE shows no degradation in generalization and exhibits no evident model drift, demonstrating that the proposed decoupling of personalization and generalization is indeed effective.

## 4.6 SCALABILITY ANALYSIS (RQ5)

To evaluate the scalability of our approach, we conducted experiments testing SOTA methods across varying numbers of clients. As shown in Table 4, our approach outperformed other methods, particularly on the CIFAR-100 and Tiny-ImageNet datasets. In scenarios with 100 clients, it achieved remarkable accuracy improvements of 18% and 48%, respectively. On CIFAR-100, increasing clients shrinks each local set to about 500 images. Consequently, PEG experts begin to over-fit, narrowing our lead to 18%. On Tiny-ImageNet, the richer data let the elastic SE continuously spawn new specialists, widening the gap to 48%. Thanks to the elastic strategy, our approach can effectively handle scenarios with a large number of clients while maintaining generalized performance, which is crucial for real-world applications.

Table 4: Test accuracy(%) with different client numbers on datasets in HtFL scenarios.

| Dataset | CIFAR-100 | | | Tiny-Imagenet | | |
|---|---|---|---|---|---|---|
| K | 20 Clients | 50 Clients | 100 Clients | 20 Clients | 50 Clients | 100 Clients |
| FedKD | 33.62 | 31.19 | 29.41 | 18.24 | 19.38 | 17.44 |
| FedProto | 29.12 | 25.13 | 23.11 | 6.93 | 2.57 | 1.51 |
| FedMRL | 39.19 | 36.01 | 35.52 | 18.74 | 19.38 | 17.21 |
| FedTGP | 35.64 | 31.96 | 35.19 | 12.61 | 12.21 | 11.33 |
| FedSA | 39.41 | 33.27 | 31.65 | 28.18 | 21.27 | 17.43 |
| **Ours** | **49.03(↑24.41%)** | **47.69(↑32.43%)** | **41.76(↑18.67%)** | **29.12(↑3.33%)** | **29.91(↑40.62%)** | **25.82(↑48.05%)** |

## 4.7 COMMUNICATION OVERHEAD ANALYSIS

Figure 6 shows FedEMoE's communication overhead compared to model-based federated learning methods. Despite the expanding shared-expert mechanism, our peak consumption remains lower than all baselines. Traffic stops increasing once the model converges, confirming the effectiveness of our dynamic-threshold adjustment strategy.

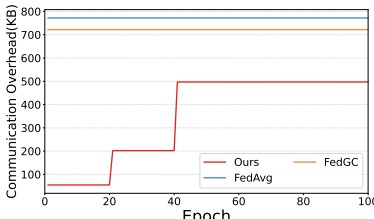

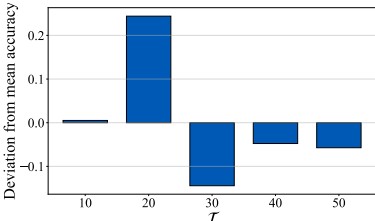

Figure 6: The communication overhead of different methods.

Figure 7: The deviation from average accuracy on different intervals of elastic operation.

## 4.8 SENSITIVITY ANALYSIS

To investigate the impact of the interval for shared expert scaling operations on the server, we evaluate model accuracy under Dirichlet $\beta = 0.1$. As illustrated in Figure 7, an interval of 20 iterations yields the best performance. We attribute this phenomenon to two complementary factors. First, overly frequent ($\mathcal{T} = 10$) scaling operations continuously perturb the model architecture, impeding sufficient convergence. Conversely, excessively sparse ($\mathcal{T} = 30, 40, 50$) operations curtail structural adaptation, preventing the shared experts from evolving toward an optimal configuration. We provide more sensitivity experiments in the Appendix F.4.

In Table 5 we co-vary the number of shared experts N (3, 4, 5) and the gate sparsity top-K (1, 2, 3). All runs use $\mathcal{T} = 2$. The best result (49.03%) is again obtained with N = 3 and top-K = 2. Adding more experts or activating more of them increases communication consumption and encourages each expert to memorize tiny client-specific patterns, which hurts generalization. Restricting the gate to top-K = 1, on the other hand, under-uses the ensemble and drops accuracy by roughly 0.4%. The small spread (less than 0.8%) indicates robustness to this design choice.

Table 5: Joint sensitivity to expert count($N$) and topK.

| topK \ experts | 3 | 4 | 5 |
|---:|:---:|:---:|:---:|
| 1 | 49.60 | 47.46 | 46.99 |
| 2 | **49.03** | 47.33 | 46.88 |
| 3 | 48.80 | 48.62 | 46.31 |

Table 6 tests the expansion trigger. We sweep the rank threshold $v_0 \in (0.75, 0.85, 0.95)$ and the exponential-smoothing factor $\gamma \in (0.05, 0.07, 0.10)$, keeping $\mathcal{T} = 2$, N = 3 and $topK = 2$ fixed. The pair ($v_0 = 0.85$, $\gamma = 0.07$) gives the highest accuracy (49.03%). A lower threshold splits experts too early and bloats the model, whereas a higher one delays growth and leaves under-fitting unchanged. $\gamma = 0.07$ strikes a balance: it smooths mini-batch noise yet still reacts within a few rounds. The accuracy variation across the nine settings is below 1%, confirming mild sensitivity.

Table 6: Sensitivity to expansion threshold $v_0$ and EMA factor $\gamma$.

| $\gamma \backslash v_0$ | 0.75 | 0.85 | 0.95 |
|---:|:---:|:---:|:---:|
| 0.05 | 48.11 | 48.32 | 48.35 |
| 0.07 | 48.28 | **49.03** | 47.26 |
| 0.10 | 48.40 | 49.09 | 47.61 |

## 4.9 ABLATION STUDIES

Table 7: Ablation studies on the key components of FedEMoE on CIFAR-100 with K = 50.

| Variant | Multi-Scale Features (PEG) | EMoE architecture (SE) | Knowledge Exchange | Accuracy(%) |
|:---:|:---:|:---:|:---:|:---:|
| Ours | ✔ | ✔ | ✔ | 47.69 |
| Ours(w/o Knowledge Exchange) | ✔ | ✔ | ✗ | 43.85(-3.84) |
| Ours(w/o PEG) | ✗ | ✔ | ✔ | 35.46(-12.23) |
| Ours(w/o SE) | ✔ | ✗ | ✔ | 42.25(-5.44) |

In the Table 7, the results show that activating all three components: personalized experts (PEG), shared experts (SE) with EMoE architecture, and knowledge exchange, yields 47.69% accuracy. Disabling knowledge exchange drops the score to 43.85%, confirming that multi-scale feature is indispensable. Retaining only PEG and knowledge exchange while removing SE's EMoE architecture causes a sharp decline to 35.46%, underscoring the role of SE in transferring knowledge and flexible fitting. Conversely, keeping SE and knowledge exchange but omitting PEG's multi-scale extraction achieves 42.25%, highlighting the critical contribution of PEG. Collectively, the three modules are all essential for optimal performance.

## 5 CONCLUSION

In this paper, we propose FedEMoE to decouple personalization from generalization on HtFL. FedEMoE designs a multi-scale knowledge extraction and a knowledge exchange mechanism to break the bottleneck of knowledge transfer. Moreover, FedEMoE first employs an EMoE model architecture to highlight multiple personalized knowledge. Extensive experiments show that FedEMoE surpasses existing HtFL methods under diverse statistical and model heterogeneity settings.

## ETHICS STATEMENT

All datasets used in this work are publicly available; no personal identifying information is involved. The study was conducted in accordance with institutional guidelines for responsible research and reproducible science.

## REPRODUCIBILITY STATEMENT

To facilitate reproducibility, we provide the following resources. (i) Source code: An anonymized Python implementation of the proposed model, training scripts, and all evaluation pipelines are included in the supplementary materials (code.zip). (ii) Data: The datasets used are publicly available and the data preprocessing scripts are provided in code.zip. (iii) Hyper-parameters: All hyper-parameters, random seeds, and hardware configurations are listed in Section 4.1 and Appendix E. (iv) Formal statements and complete proofs of Theorems-are given in Appendix D.

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

## A  THE USE OF LARGE LANGUAGE MODELS

In the paper writing stage of this study, we only utilized Large Language Models (LLMs) as a general language polishing tool. Specifically, we employed LLMs to conduct spelling checks, grammatical corrections, and word optimization on certain sentences in the text, aiming to enhance the clarity and fluency of language expression. LLMs did not participate in the proposal of research questions,

experimental design, data analysis, result interpretation, chart production, or the generation of any scientific content. Therefore, the role of LLMs in this work is limited to assisting in language-level polishing and does not constitute an academic contributor.

# B  NOTATIONS

| Symbol | Description |
|---|---|
| $K$ | Total number of clients. |
| $i$ | The client number. |
| $p$ | client participation fraction. |
| $T$ | Total global training rounds. |
| $\mathcal{T}$ | shared expert scaling interval. |
| $D_i$ | The client $i$'s private dataset. |
| $\mathcal{D}_i$ | The client $i$'s training dataset. |
| $M_i$ | The model parameter of the $i$-th client. |
| $C$ | Number of classes. |
| $\mathcal{N}_i$ | The number of experts in $M_i$. |
| $N$ | The number of experts in shared expert. |
| Top $k$ | The number of actived experts. |
| $FE_i$ | The feature extractor of client $i$. |
| $SE_i$ | The shared expert of client $i$. |
| $PEG_i$ | The personalized expert group of client $i$. |
| $H_i$ | The personalized header of client $i$. |
| $f_i(\cdot)$ | $FE_i$'s parameter. |
| $p_i(\cdot)$ | $PEG_i$'s parameter. |
| $s_i(\cdot)$ | $SE_i$'s parameter. |
| $h_i(\cdot)$ | $H_i$'s parameter. |

Table 8: Notations

# C  ALGORITHM

---

**Algorithm 1** FedEMoE

---

**Input:** Total number of clients $K$, Client participation fraction $J$, Initial client models $W_i^0 \,_{i=1}^K$, Initial global model $W_g^0$, Communication rounds $T$
1: **for** $t = 1$ to $T$ **do**
2:     Select a subset of clients $K_t \subseteq (1, \ldots, K)$, $|K_t| = max(1, JK)$
3:     **for** $i \in K_t$ **do**
4:         $f_i^t, s_i^t \leftarrow$ ClientUpdate($f_g^{t-1}, s_g^{t-1}, M_i^{t-1}, \mathcal{D}_i$)
5:     **end for**
6:     $f_i^t, s_i^t, B_g \leftarrow$ Weighted Aggregation($f_i^t|_{i=1}^{K_t}, s_i^t|_{i=1}^{K_t}, |\mathcal{D}_i|_{i=1}^{K_t}, B_i^t|_{i=1}^{K_t}$)
7:     Every $\mathcal{T}$ do:
8:     Parallel processing of ExpandExpert($e, p, N$) and ShrinkMoE($N, B_g, E$)
9: **end for**
10: **return** $f_i^t, s_i^t$

---

# D  THEORETICAL ANALYSIS

## D.1  THE EXPERT INITIALIZATION IN EXTENSION.

When migrating expert network $e_j$ from the original MoE to a larger-scale new MoE $e^*$, we can simply keep the $j$-th expert unchanged by setting $e_j^{new} = e_j^{old}$, while initializing all remaining positions $i \neq j$, where $j$ is the index of $e_j$ in the old MoE $W_g^{old}$. This design is justified by two key

**Algorithm 2** ClientUpdate

**Input:** Global feature extractor $f_g^{t-1}$, Global shared expert $s_g^{t-1}$, Client model $M_i^{t-1}$, Local training data $\mathcal{D}_i$
**Output:** Feature extractor $f_i^t$, Shared expert $s_i^t$
 1: $M_i^{t-1} \leftarrow \text{UpdateLocalModel}(f_g^{t-1}, s_g^{t-1}, M_i^{t-1})$
 2: DoubleKD $(M_i^{t-1}, \mathcal{D}_i)$ via equation 3 and equation 4
 3: $M_i^t \leftarrow \text{LocalTrain}(M_i^{t-1}, \mathcal{D}_i)$
 4: **return** $f_i^t, s_i^t$

---

**Algorithm 3** ExpandExpert

**Input:** Expanding expert $e$, the router $R$ of $e$'s parent-MoE, the index $p$ of $e$ in parent-MoE, number of experts $N$
**Output:** MoE-based Expert $e^* = (R, \mathcal{E})$, where $\mathcal{E} = (e_1, \ldots, e_N)$
 1: $\mathcal{E} \leftarrow InitalizeList(N)$
 2: **for** $r = 1$ **to** $N$ **do**
 3:    **if** $r = p$ **then**
 4:       $e_r \leftarrow E$
 5:    **else**
 6:       $e_r \leftarrow InitializeExpert$
 7:    **end if**
 8:    $\mathcal{E}[r] \leftarrow e_r$
 9: **end for**
10: $e^* \leftarrow (R, \mathcal{E})$
11: **return** $e^*$

---

**Algorithm 4** ShrinkMoE

**Input:** Number of experts $N$, Activation proportion vector $B_g = (b_{g,1}, b_{g,2}, \ldots, b_{g,N})^\top$, MoE-based Expert $e^* = (e_1, e_2, \ldots, e_N)^\top$.
**Output:** Shrunk expert $e$
 1: $e \leftarrow \mathbf{0}$
 2: **for** $r = 1$ **to** $N$ **do**
 3:    $e \leftarrow e + b_{g,r} \cdot e_r$
 4: **end for**
 5: **return** $e$

properties. First, the routing distribution remains invariant: the original gate's preference for the $j$-th expert is expressed as $p_j(x) = exp(h^T w_j^{old}) / \sum_k exp(h^T w_k^{old})$. By this, we ensure $p_j^{new}(x) \approx p_j^{old}(x)$ for any input $x$, thereby preventing catastrophic forgetting. Second, the overall output of the new MoE can be written as $y^{new}(x) = G^{new}(x)j e_j^{old}(x) + \sum_{i \neq j} G^{new}(x)_i e_i^{new}(x)$. The first term preserves existing capability, while the second term, thanks to the random initialization of $E_i^{new}$, forms an approximately orthogonal basis that introduces additional representational freedom. During training, the gate will sparsely assign new tokens to these fresh experts, delivering a provably incremental expansion of model capacity without erasing prior knowledge.

## D.2 GLOBAL CHECK VIA SINGULAR SPECTRUM

**Theorem 1** *Let round $t$ proceed solely by server-side weighted aggregation of the uploaded local models $M_i$ into a global model $M^{(t)} = \sum_{i=1}^{p} w_i M_i$, with $\sum_i w_i = 1$, $w_i \geq 0$. Define the per-round client covariance $\Sigma = \sum_i w_i(M_i - M)(M_i - M)^T$. Then, for every $k$, with probability at least $1 - \frac{1}{poly(d)}$, we have:*

$$|\sigma_k(M^{(t)}) - \sigma_k(M)| \leq \sqrt{\frac{||\Sigma||_2 log d}{p}} + \frac{L log d}{3p}, \quad (9)$$

*where $\sigma_k(\cdot)$ denotes the $k$-th largest singular value, $L$ is the Lipschitz constant of the model parameters, and $p$ is the number of participating clients, $poly(d)$ is a notation emphasizing that the probability approaches $1$ exponentially as the dimension $d$ increases.*

**Remark 1**. The Theorem 1 justifies using the aggregated model's weights as a proxy for the training status of all participating models. If the singular spectrum of $M^{(t)}$ decays rapidly (light tail), $||\Sigma||_2$ must be a small value, indicating negligible drift and confirming that the aggregated model is already close to the ideal drift-free model $M_i$. Conversely, a heavy tail or abrupt large singular values directly reveal substantial drift, flagging the model as unreasonable. Thus, the aggregated weight matrix alone suffices for a theoretically grounded validity check.

## D.3 CONVERGENCE ANALYSIS FOR STRONGLY CONVEX CASE

or non-convex case In this subsection, we provide a formal convergence guarantee for the proposed FedEMoE approach under standard assumptions commonly used in federated optimization.

### D.3.1 ASSUMPTION

**Assumption 1 (Smoothness)** *$F$ is L-smooth: for all vector $\theta_1$ and $\theta_2$,*

$$\|\nabla F(\theta_1) - \nabla F(\theta_2)\| \leq L\|\theta_1 - \theta_2\|.$$

*Another form:*

$$F(\theta_1) \leq F(\theta_2) + <\theta_1 - \theta_2, \nabla F(\theta_2)> + \frac{L}{2}\|\theta_1 - \theta_2\|^2.$$

**Assumption 2 (Convexity)** *Assume $F$ is strongly convex with parameter $\mu$. For any vector $\theta_1$, $\theta_2$, we have:*

$$F(\theta_1) \geq F(\theta_2) + <\theta_1 - \theta_2, \nabla F(\theta_2)> + \frac{\mu}{2}\|\theta_1 - \theta_2\|^2.$$

**Assumption 3 (Stochastic Gradient Variance Bounded)**

$$E[\|\nabla F(\theta; x, y) - \nabla F(\theta)^2] \leq \sigma^2.$$

**Assumption 4 (MoE Boundedness)** *The gating weights $\{\pi_j\}$ satisfy $\sum_{j=1}^{k} \pi_j$ and each expert's outputs and gradients are uniformly bounded.*

### D.3.2 KEY LEMMAS

**Lemma 1** *Assume Assumption 1, 2, 3 hold, if local learning rate $\eta_l \leq \frac{1}{L}$. Then the local update on the client side satisfies:*

$$E||\theta_{i,E}^t|| \leq (1 - \eta_l\mu)^e||\theta^t - \theta^*|| + \frac{2\eta_l\sigma^2}{\eta}. \tag{10}$$

**Lemma 2** *Assume Assumption 1, 2 hold. Then the aggregation on the server satisfies:*

$$E||\theta^{t+1} - \theta^*||^2 \leq \sum_i w_i E||\theta_{i,e}^t - \theta^*||^2. \tag{11}$$

**Lemma 3** *Set linear projection $P$ satisfies $||p|| \leq 1$ then*

$$||P(x) - \theta^*|| \leq ||x - \theta^*||. \tag{12}$$

### D.3.3 PROOF OF KEY LEMMA

**Proof 1** *(Proof of Lemma 1) For any local step $\gamma$ Taking squared norms and expectations,*

$$\Delta_{\gamma+1} = \Delta_\gamma - \eta_l\nabla f(\theta_{i,\gamma}^t; \xi_{i,\gamma}^t). \tag{13}$$

*Taking squared norms and expectations, we can get*

$$E||\Delta_{\gamma+1}||^2 \tag{14}$$
$$= E||\Delta_\gamma - \eta_l\nabla f(\theta_{i,\gamma}^t; \xi_{i,\gamma}^t)||^2.$$
$$= E[||\Delta_\gamma||^2 - 2\eta_l < \Delta_\gamma, \nabla f(\theta_{i,\gamma}^t; \xi_{i,\gamma}^t > +\eta_l^2||\nabla f(\theta_{i,\gamma}^t; \xi_{i,\gamma}^t||^2]$$

*Substituting assumption 3 yields:*

$$E||\Delta_{\gamma+1}||^2 \leq ||\Delta_\gamma||^2 - 2\eta_l < \Delta_\gamma, \nabla F(\theta_{i,\gamma}^t) > +\eta_l^2(||\nabla F(\theta_{i,\gamma}^t)||^2 + \sigma^2) \tag{15}$$

*Substituting assumption 2,1 yields:*

$$E||\Delta_{\gamma+1}||^2 \leq (1 - \eta_l\mu)||\Delta_\gamma||^2 + \eta_l^2\sigma^2 \tag{16}$$

*Unrolling the recursion for $\sigma = 0, 1, \ldots, e-1$:*

$$E||\Delta_e||^2 \leq (1 - \mu\eta_l)^e||\Delta_0||^2 + \eta_l^2\sigma^2 \sum_{\gamma=0}^{e-1}(1 - \mu\eta_l)^\gamma. \tag{17}$$

*Therefore,*

$$E||\Delta_e||^2 \leq (1 - \mu\eta_l)^e||\theta^t - \theta^*||^2 + \frac{\eta_l\sigma^2}{\mu}. \tag{18}$$

*To match the statement in Lemma 1, we further loosen the constant term to absorb higher-order effects:*

$$E||\theta_{i,E}^t|| \leq (1 - \eta_l\mu)^e||\theta^t - \theta^*|| + \frac{2\eta_l\sigma^2}{\eta}. \tag{19}$$

**Proof 2** *(Proof of Lemma 2) From Lemma 1 (already proved) we have for every client $i$:*

$$E||\theta_{i,E}^t - \theta^*||^2 \leq (1 - \mu\eta_l)^e||\theta_g^t - \theta^*||^2 + \frac{2\eta_l\sigma^2}{\mu}. \tag{20}$$

*According to Jensen's inequality (convex combination), we can get:*

$$E\left\|\sum_{i=1}^N w_i\theta_{i,e}^t - \theta^*\right\|^2 \tag{21}$$

$$\leq \sum_{i=1}^N w_i E||\theta_{i,e}^t - \theta^*||^2$$

$$\leq (1 - \mu\eta_l)^e||\theta_g^t - \theta^*||^2 + \frac{2\eta_l\sigma^2}{\mu}.$$

*Because $P$ is non-expansive and $\theta^*$ is in the fixed-point set of $P$*

$$\|\theta_g^{t+1} - \theta^*\| = \|P(\bar{\theta}^{t+1}) - \theta^*\| \leq \|\bar{\theta}^{t+1} - \theta^*\|, \tag{22}$$

*where $\bar{\theta}^{t+1} = \sum_i w_i \theta_{i,e}^t$. Substituting assumption 2,1 yields:*

$$F(\theta_{virt}) \leq F(\theta_g^t) - \eta_g \left(1 - \frac{L\eta_g}{2}\right) 2\mu\big(F(\theta_g^t) - F(\theta^*)\big). \tag{23}$$

*With $\eta_g \leq 2\mu^2/\beta^3$ (choose $\beta = 2L$ for simplicity),so*

$$F(\theta_{virt}) \leq F(\theta_g^t) - \mu\eta_g\big(F(\theta_g^t) - F(\theta^*)\big) + \frac{\eta_g^2\beta\sigma^2}{2}. \tag{24}$$

*Collecting all residual terms and using the previous bounds, we obtain:*

$$E[L_a(\theta_g^{t+1})] \leq E[L_a(\theta_g^t)] - \frac{\mu^2}{2}\eta_g E\|\theta_g^t - \theta^*\|^2 + \frac{\eta_g^2\beta\sigma^2}{2}. \tag{25}$$

**Proof 3** *(Proof of Lemma 3) Define the error vector $\Delta_t = \tilde{\theta}_i^t - \tilde{\theta}_i^*$.. The update gives $\Delta_{t+1} = \Delta_t - \rho\nabla f_i(\tilde{\theta}_i^t; \xi_i^t)$. Taking squared norms and expectations:*

$$E\|\Delta_{t+1}\|^2 = E\|\Delta_t - \rho\nabla f_i(\tilde{\theta}_i^t; \xi_i^t)\|^2 \tag{26}$$

$$= E\Big[\|\Delta_t\|^2 - 2\rho\langle\Delta_t, \nabla f_i(\tilde{\theta}_i^t; \xi_i^t)\rangle + \rho^2\|\nabla f_i(\tilde{\theta}_i^t; \xi_i^t)\|^2\Big].$$

*Substituting assumption 2,1 yields:*

$$E\|\Delta_{t+1}\|^2 \leq \|\Delta_t\|^2 + 2\rho\Big(F_i(\tilde{\theta}_i^t) - F_i(\tilde{\theta}_i^*) - \frac{\mu}{2}\|\Delta_t\|^2\Big) + \rho^2\Big(\gamma\big(F_i(\tilde{\theta}_i^t) - F_i(\tilde{\theta}_i^*)\big) + \sigma^2\Big) \tag{27}$$

$$= (1 - \mu\rho)\|\Delta_t\|^2 + \big(2\rho + \rho^2\gamma\big)\big(F_i(\tilde{\theta}_i^t) - F_i(\tilde{\theta}_i^*)\big) + \rho^2\sigma^2.$$

*Because $\rho \leq 2/\gamma$ implies $2\rho + \rho^2\gamma \leq 0$, and $F_i(\ldots) - F_i(\ldots^*) \leq 0$, we obtain:*

$$E\|\Delta_{t+1}\|^2 \leq (1 - \mu\rho)\|\Delta_t\|^2 + \rho^2\sigma^2. \tag{28}$$

*Substituting assumption 3 yields:*

$$E[L_p(\tilde{\theta}_i^{t+1})] \leq E[L_p(\tilde{\theta}_i^t)] - \mu^2\rho E\|\tilde{\theta}_i^t - \tilde{\theta}_i^*\|^2 + \frac{\gamma\rho^2\sigma^2}{2}. \tag{29}$$

*Lemma 3 is proved, showing that each personalized-parameter update yields a linear decrease in the expected personalized loss, up to a variance term controlled by rho*

### D.3.4 MAIN THEOREM

**Theorem 2** *Suppose Assumptions 1,2,3 and 4 hold. Let the local learning rate satisfy $\eta_l \leq \min\left\{\frac{1}{L}, \frac{\log(e)}{e\mu}\right\}$,. Then:*

$$E[F(\theta_{final}^T) - F(\theta^*)] \leq \frac{2L\sigma^2}{\mu^2 eT} + \frac{\mu}{2}\exp(-\mu\eta_l eT)\|\theta^0 - \theta^*\|^2. \tag{30}$$

*According to Lemma 1,2,3, we can get:*

$$\sum_{k=0}^{T-1}(1 - \mu\eta_l)^{ek} \leq \sum_{k=0}^{\infty}(1 - \mu\eta_l)^{ek} = \frac{1}{1 - (1 - \mu\eta_l)^e} \leq \frac{1}{\mu\eta_l e}, \tag{31}$$

*because $(1 - x)^e \leq 1 - ex$ for $0 < x < 1/e$, hence $D_T \leq (1 - \mu\eta_l)^{eT}D_0 + \frac{2\sigma^2}{\mu^2 e}$. By strong convexity, $F(\theta) - F(\theta^*) \leq \frac{L}{2}\|\theta - \theta^*\|^2$. Apply to $\theta = \theta_{final}^T = P(\theta_g^T)$:*

$$E[F(\theta_{final}^T) - F(\theta^*)] \leq \frac{L}{2}E\|\theta_g^T - \theta^*\|^2 \leq \frac{L}{2}\left[(1 - \mu\eta_l)^{eT}D_0 + \frac{2\sigma^2}{\mu^2 e}\right]. \tag{32}$$

*Take $\eta_l = \frac{\log(T)}{e\mu T}$. Then*

$$(1 - \mu\eta_l)^{eT} \leq \exp(-\mu\eta_l eT) = \exp(-\log T) = \frac{1}{T}. \tag{33}$$

*Thus:*

$$E[F(\theta_{final}^T) - F(\theta^*)] \leq \frac{L}{2}\left[\frac{D_0}{T} + \frac{2\sigma^2}{\mu^2 e}\right]. \tag{34}$$

*FedEMoE converges at a rate $\mathcal{O}\big(\frac{1}{T} + \frac{1}{e}\big)$, showing linear speed-up with the number of local steps $e$ and communication rounds $T$.*

## D.4 Convergence Analysis for non-convex case

**Assumption 5 (Smoothness)** *$F$ is $L$-smooth: for all vector $\theta_1$ and $\theta_2$,*

$$\|\nabla F(\theta_1) - \nabla F(\theta_2)\| \leq L\|\theta_1 - \theta_2\|. \tag{35}$$

**Assumption 6 (Stochastic Gradient Variance Bounded)**

$$E[\|\nabla F(\theta; x, y) - \nabla F(\theta)^2] \leq \sigma^2. \tag{36}$$

**Assumption 7 (MoE Boundedness)**

$$|\mathbf{e}_j(x)| \leq G, \ |\nabla \mathbf{e}_j(x)| \leq L_e \tag{37}$$

### D.4.1 Key Lemma

**Lemma 4** *If the local learning rate satisfies $\eta_l \leq \frac{1}{2L}$, then:*

$$\frac{1}{N}\sum_{i=1}^{N} E\|x_{i,e}^t - x^t\|^2 \leq 4e^2\eta_l^2\|\nabla F(x^t)\|^2 + 2e\eta_l^2\sigma^2. \tag{38}$$

**Lemma 5** *For a non-expansive linear projection $P$ with operator norm $\|P\| \leq 1$, the aggregated update satisfies:*

$$E\|x^{t+1} - x^t\|^2 \leq \frac{1}{N}\sum_{i=1}^{N} E\|x_{i,e}^t - x^t\|^2. \tag{39}$$

**Lemma 6** *If the local step size satisfies $\eta_l \leq \frac{1}{2L}$, then:*

$$E[F(x^{t+1})] \leq E[F(x^t)] - \frac{e\eta_l}{4}E\|\nabla F(x^t)\|^2 + \frac{Le\eta_l^2\sigma^2}{2}. \tag{40}$$

### D.4.2 Proof of key lemma

**Proof 4** *(Proof of Lemma 4) Take squared norms and expectations:*

$$E\|\delta_{i,\tau+1}\|^2 = E\|\delta_{i,\tau} - \eta_l g_{i,\tau}^t\|^2 = E\|\delta_{i,\tau}\|^2 - 2\eta_l E\langle\delta_{i,\tau}, g_{i,\tau}^t\rangle + \eta_l^2 E\|g_{i,\tau}^t\|^2. \tag{41}$$

*Substituting assumption 6,5 yields:*

$$E\|g_{i,\tau}^t\|^2 \leq 2\|\nabla F_i(x^t)\|^2 + 2L^2\|\delta_{i,\tau}\|^2 + \sigma^2. \tag{42}$$

*Insert the bound, we can get:*

$$E\|\delta_{i,\tau+1}\|^2 \leq E\|\delta_{i,\tau}\|^2 - 2\eta_l E\langle\delta_{i,\tau}, \nabla F_i(x^t)\rangle + \eta_l^2\left[2\|\nabla F_i(x^t)\|^2 + 2L^2\|\delta_{i,\tau}\|^2 + \sigma^2\right]. \tag{43}$$

*Then:*

$$-2\eta_l E\langle\delta_{i,\tau}, \nabla F_i(x^t)\rangle \leq 2\eta_l^2\|\nabla F_i(x^t)\|^2 + \frac{1}{2}\|\delta_{i,\tau}\|^2. \tag{44}$$

*Unrolling over $\tau = 0, \ldots, e-1$ gives:*

$$E\|\delta_{i,e}\|^2 \leq 4e^2\eta_l^2\|\nabla F_i(x^t)\|^2 + 2e\eta_l^2\sigma^2. \tag{45}$$

*Taking the average over all clients:*

$$\frac{1}{N}\sum_{i=1}^{N} E\|\delta_{i,e}\|^2 \leq 4e^2\eta_l^2\|\nabla F(x^t)\|^2 + 2e\eta_l^2\sigma^2, \tag{46}$$

**Proof 5** *(Proof od Lemma 5) The squared norm is convex, so by Jensen's inequality* $\bar{x}^{t+1} = \sum_{i=1}^{N} w_i x_{i,e}^t$. *The squared norm is convex, so by Jensen's inequality:*

$$\|\bar{x}^{t+1} - x^t\|^2 = \left\|\sum_{i=1}^{N} w_i(x_{i,e}^t - x^t)\right\|^2 \leq \sum_{i=1}^{N} w_i \|x_{i,e}^t - x^t\|^2. \tag{47}$$

*Taking expectation over the randomness yields*

$$E\|\bar{x}^{t+1} - x^t\|^2 \leq \sum_{i=1}^{N} w_i E\|x_{i,e}^t - x^t\|^2 = \frac{1}{N}\sum_{i=1}^{N} E\|x_{i,e}^t - x^t\|^2. \tag{48}$$

*The final update is* $x^{t+1} = P(\bar{x}^{t+1})$, *where the linear operator* $P$ *satisfies* $\|P(u) - P(v)\| \leq \|u - v\|, \quad \forall u, v$. *Hence:*

$$\|x^{t+1} - x^t\|^2 = \|P(\bar{x}^{t+1}) - P(x^t)\|^2 \leq \|\bar{x}^{t+1} - x^t\|^2. \tag{49}$$

*Combining the above gives the desired bound:*

$$E\|x^{t+1} - x^t\|^2 \leq \frac{1}{N}\sum_{i=1}^{N} E\|x_{i,e}^t - x^t\|^2. \tag{50}$$

**Proof 6** *(Proof of Lemma 6) By Assumption 5, for any two points we have* $F(y) \leq F(x) + \langle \nabla F(x), y - x \rangle + \frac{L}{2}\|y - x\|^2$. *Set* $x = x^t, \quad y = x^{t+1}$. *Then:*

$$E[F(x^{t+1})] \leq E[F(x^t)] + E\langle \nabla F(x^t), x^{t+1} - x^t \rangle + \frac{L}{2}E\|x^{t+1} - x^t\|^2. \tag{51}$$

*From Lemma 5 we have:*

$$E\|x^{t+1} - x^t\|^2 \leq \frac{1}{N}\sum_{i=1}^{N} E\|x_{i,e}^t - x^t\|^2. \tag{52}$$

*Write the inner product term explicitly:*

$$E\langle \nabla F(x^t), x^{t+1} - x^t \rangle = E\left\langle \nabla F(x^t), \sum_{i=1}^{N} w_i(x_{i,e}^t - x^t)\right\rangle = \sum_{i=1}^{N} w_i E\langle \nabla F(x^t), x_{i,e}^t - x^t \rangle. \tag{53}$$

*For each client* $i$ *and step* $\tau$ *we have:*

$$x_{i,e}^t - x^t = -\eta_l \sum_{\tau=0}^{e-1} g_{i,\tau}^t, \tag{54}$$

*so we can get:*

$$E\langle \nabla F(x^t), x_{i,e}^t - x^t \rangle \tag{55}$$

$$= -e\eta_l \langle \nabla F(x^t), \nabla F_i(x^t) \rangle \tag{56}$$

$$\leq -e\eta_l \|\nabla F(x^t)\|^2 + 2Le^2\eta_l^3 \|\nabla F(x^t)\|^2 + e\eta_l^2\sigma^2. \tag{57}$$

*Insert the bounds from all above:*

$$E[F(x^{t+1})] \leq E[F(x^t)] - e\eta_l \|\nabla F(x^t)\|^2 + 2Le^2\eta_l^3 \|\nabla F(x^t)\| \tag{58}$$

$$+ e\eta_l^2\sigma^2 + \frac{L}{2}\left(4e^2\eta_l^2 \|\nabla F(x^t)\|^2 + 2e\eta_l^2\sigma^2\right).$$

*When* $\eta_l \leq 1/(2L)$ *we have* $2Le^2\eta_l^3 + 2Le^2\eta_l^2 \leq \frac{e\eta_l}{4}$, *and* $e\eta_l^2\sigma^2 + Le\eta_l^2\sigma^2 \leq \frac{Le\eta_l^2\sigma^2}{2}$. *Hence:*

$$E[F(x^{t+1})] \leq E[F(x^t)] - \frac{e\eta_l}{4}E\|\nabla F(x^t)\|^2 + \frac{Le\eta_l^2\sigma^2}{2}. \tag{59}$$

### D.4.3 MAIN THEOREM FOR NON-CONVEX CASE

**Theorem 3** *Choose the local learning rate $\eta_l = \frac{1}{\sqrt{eTL}}$. Then after $T$ communication rounds,*

$$\frac{1}{T} \sum_{t=0}^{T-1} E\|\nabla F(x^t)\|^2 \leq \frac{2L\Delta_0}{\sqrt{eT}} + \frac{\sigma^2}{\sqrt{eT}}, \tag{60}$$

*We have for each round $t$:*

$$E[F(x^{t+1})] \leq E[F(x^t)] - \frac{e\eta_l}{4} E\|\nabla F(x^t)\|^2 + \frac{Le\eta_l^2\sigma^2}{2}. \tag{61}$$

*Move the gradient term to the left, we can get:*

$$\frac{e\eta_l}{4} E\|\nabla F(x^t)\|^2 \leq E[F(x^t)] - E[F(x^{t+1})] + \frac{Le\eta_l^2\sigma^2}{2}. \tag{62}$$

*Sum the above inequality over $t = 0, 1, \ldots, T-1$:*

$$\sum_{t=0}^{T-1} \frac{e\eta_l}{4} E\|\nabla F(x^t)\|^2 \leq F(x^0) - E[F(x^T)] + \frac{Le\eta_l^2\sigma^2 T}{2}. \tag{63}$$

*Because $F(x^0) - E[F(x^T)] \leq F(x^0) - F = \Delta_0$. Hence:*

$$\sum_{t=0}^{T-1} \frac{e\eta_l}{4} E\|\nabla F(x^t)\|^2 \leq \Delta_0 + \frac{Le\eta_l^2\sigma^2 T}{2}. \tag{64}$$

*Insert $\eta_l = \frac{1}{\sqrt{eTL}}$, we can get:*

$$\frac{\sqrt{e}}{4\sqrt{TL}} \sum_{t=0}^{T-1} E\|\nabla F(x^t)\|^2 \leq \Delta_0 + \frac{\sigma^2}{2}. \tag{65}$$

*Then:*

$$\frac{1}{T} \sum_{t=0}^{T-1} E\|\nabla F(x^t)\|^2 \leq \frac{4\sqrt{TL}}{\sqrt{eT}}\Delta_0 + \frac{4\sqrt{TL}}{\sqrt{eT}}\frac{\sigma^2}{2} = \frac{4\sqrt{L}}{\sqrt{eT}}\Delta_0 + \frac{2\sqrt{L}}{\sqrt{eT}}\sigma^2. \tag{66}$$

*Under non-convexity, FedEMoE achieves $\mathcal{O}\left(\frac{1}{\sqrt{eT}}\right)$ convergence to a stationary point, linearly dependent on the product of local steps $e$ and communication rounds $T$.*

### D.5 PROOF OF CONVERGENCE OF EXPANSION AND SHRINKAGE

### D.5.1 EXPANSION DOES NOT INCREASE THE LIPSCHITZ CONSTANT

**Lemma 7** *Let the original model parameter be $\theta \in R^d$, and after applying the **expansion** operation, the new parameter becomes:*

$$\theta' = (e_1, e_2, \ldots, e_N) \in R^{N \cdot d} \quad \text{with} \quad e_p = e, \quad \text{and other } e_i \text{ initialized arbitrarily.}$$

*We assume the loss function $F_k(\theta)$ is **Lipschitz continuous** in $\theta$, i.e.,*

$$\|\nabla F_k(\theta_1) - \nabla F_k(\theta_2)\| \leq L\|\theta_1 - \theta_2\|$$

**Proof 7** *Let us denote:*

$$\theta'_1 = (e_{1,1}, \ldots, e_{1,N}), \quad \theta'_2 = (e_{2,1}, \ldots, e_{2,N}) \tag{67}$$

*Since **expansion** only introduces new parameters and does **not modify** the original expert $e$, we can write:*

$$\nabla_{\theta'} F(\theta') = (\nabla_{e_1} F(\theta'), \ldots, \nabla_{e_N} F(\theta')) \tag{68}$$

*Note that:*

- $\nabla_{e_p} F(\theta') = \nabla_e F(\theta)$

- *For $i \neq p$, $\nabla_{e_i} F(\theta') = 0$ (since new experts are not trained or initialized randomly and not used in forward pass)*

*Thus:*

$$\|\nabla F(\theta_1') - \nabla F(\theta_2')\|^2 = \sum_{i=1}^{N} \|\nabla_{e_{1,i}} F - \nabla_{e_{2,i}} F\|^2 \tag{69}$$
$$= \|\nabla_e F(\theta_1) - \nabla_e F(\theta_2)\|^2$$
$$\leq L^2 \|e_1 - e_2\|^2$$
$$\leq L^2 \|\theta_1' - \theta_2'\|^2$$

*Taking square roots:*

$$\|\nabla F(\theta_1') - \nabla F(\theta_2')\| \leq L\|\theta_1' - \theta_2'\| \tag{70}$$

### D.5.2  SHRINKAGE DOES NOT INCREASE THE LIPSCHITZ CONSTANT

**Lemma 8** *Let the MoE model be:*

$$\theta = (e_1, e_2, \ldots, e_N) \in R^{N \cdot d} \tag{71}$$

*After applying the **Shrinkage** operation, we obtain:*

$$\hat{e} = \sum_{r=1}^{N} a_r e_r, \quad \text{with } a_r \geq 0, \sum_{r=1}^{N} a_r = 1 \tag{72}$$

**Proof 8** *We first compute the gradient of $F$ with respect to the compressed expert:*

$$\nabla_{\hat{e}} F = \sum_{r=1}^{N} a_r \nabla_{e_r} F \tag{73}$$

*Then:*

$$\|\nabla F(\hat{e}_1) - \nabla F(\hat{e}_2)\| = \left\| \sum_{r=1}^{N} a_r \left( \nabla_{e_{r,1}} F - \nabla_{e_{r,2}} F \right) \right\| \tag{74}$$
$$\leq \sum_{r=1}^{N} a_r \|\nabla_{e_{r,1}} F - \nabla_{e_{r,2}} F\|$$
$$\leq \sum_{r=1}^{N} a_r L \|e_{r,1} - e_{r,2}\|$$
$$= L \sum_{r=1}^{N} a_r \|e_{r,1} - e_{r,2}\|$$

*Now, since:*

$$\|\hat{e}_1 - \hat{e}_2\| = \left\| \sum_{r=1}^{N} a_r (e_{r,1} - e_{r,2}) \right\| \leq \sum_{r=1}^{N} a_r \|e_{r,1} - e_{r,2}\| \tag{75}$$

*We conclude:*

$$\|\nabla F(\hat{e}_1) - \nabla F(\hat{e}_2)\| \leq L \sum_{r=1}^{N} a_r \|e_{r,1} - e_{r,2}\| \leq L\|\hat{e}_1 - \hat{e}_2\| \tag{76}$$

### D.5.3   SUMMARY

- **Expansion** introduces new parameters but does **not affect** the original expert or its gradients $\Rightarrow$ **Lipschitz constant unchanged**.

- **Shrinkage** is a **convex combination** of experts $\Rightarrow$ **gradient Lipschitz constant preserved**.

- Hence, **both operations preserve the convergence properties** of federated learning algorithms under standard smoothness assumptions.

### D.6   COMPLEXITY ANALYSIS

To ensure a fair comparison across all five methods, we adopt a uniform set of driving factors when deriving per-round time complexity:

- **K** – total number of clients;

- **C** – number of classes in the classification task;

- **d** – dimension of each prototype / feature vector (used for FedProto, FedTGP and FedSA);

- **P** – parameter count of the lightweight homogeneous model exchanged in FedMRL ($P \ll M$);

- **B** – local batch count per client per round;

- **E** – number of local epochs executed on each client;

- **M** – total parameter count of a client-side heterogeneous model;

- **S** – server-side training iterations required by FedTGP's contrastive step.

With these quantities we decompose the cost into three common phases: local forward-and-backward training, server aggregation (or prototype / anchor optimisation), and bidirectional communication. The resulting expressions therefore reflect the worst-case computational workload and communication volume that each algorithm imposes in a single federated round.

- **FedKD.** Each client trains its heterogeneous model and performs forward propagation on the broadcast homogeneous model to generate soft labels for distillation. Every forward/backward pass involves the full model parameters $M$, repeated for $E$ epochs and $B$ batches per epoch, giving a local compute cost of $E \cdot B \cdot M$. With $K$ clients running in parallel, the total client-side cost is $K \cdot E \cdot B \cdot M$. The server only averages the uploaded homogeneous models, yielding $O(KM)$ computation and communication, which is absorbed by the client term. Hence the per-round complexity is $O(K \cdot E \cdot B \cdot M)$.

- **FedProto.** Local training is identical to FedKD, hence $K \cdot E \cdot B \cdot M$ is retained. After training, each client computes $C$ class prototypes by averaging $d$-dimensional features; the cost $C \cdot d$ is negligible compared with $M$. The server collects and averages these prototypes from $K$ clients, requiring $O(K \cdot C \cdot d)$ computation and communication. The overall per-round complexity becomes $O(K \cdot E \cdot B \cdot M + K \cdot C \cdot d)$.

- **FedMRL.** Every client runs both its heterogeneous model and the global homogeneous small model once per sample, concatenates their representations, and feeds the result into a lightweight projector. The projector's parameter count is $\ll M$, so the local cost is still dominated by $E \cdot B \cdot M$. With $K$ clients the client-side total is $K \cdot E \cdot B \cdot M$. The server aggregates only the homogeneous small model whose size is denoted by $P \ll M$, giving $O(K \cdot P)$ computation and communication. Thus the per-round complexity is $O(K \cdot E \cdot B \cdot M + K \cdot P)$.

- **FedTGP.** Local training and prototype extraction are the same as in FedProto, contributing $K \cdot E \cdot B \cdot M + K \cdot C \cdot d$. Instead of simple averaging, the server performs $S$ iterations of contrastive learning among $C$ trainable prototypes. Each iteration computes $O(C^2)$ pairwise distances of dimension $d$, adding $S \cdot C^2 \cdot d$ server-side computation. Communication remains prototype-based, i.e., $O(K \cdot C \cdot d)$. The total per-round complexity is therefore $O(K \cdot E \cdot B \cdot M + K \cdot C \cdot d + S \cdot C^2 \cdot d)$.

- **FedSA.** Clients conduct standard model training ($E \cdot B \cdot M$) and then align features with $C$ semantic anchors of dimension $d$; the anchor-related cost $C \cdot d$ is negligible versus $M$. With $K$ clients the client-side total is $K \cdot E \cdot B \cdot M$. The server only averages the anchor vectors, yielding $O(K \cdot C \cdot d)$ computation and communication. Hence the per-round complexity is $O(K \cdot E \cdot B \cdot M + K \cdot C \cdot d)$.

- **FedEMoE.** Every client runs E epochs of B mini-batches over its heterogeneous MoE , giving a client compute cost of $K \cdot E \cdot B \cdot M$. After training, each client uploads only the sparse SE sub-experters it touched together with the full FE; these are subsets of $M$, so upstream traffic is $K \cdot M$. The server first does the usual weighted/sparse aggregation (cost absorbed in the upload). It then refines each of the $N$ SE sub-experts (width $d$) by one SVD, adding $Nd^3$ server-side computation. Finally the refined global model is broadcast back, contributing another $K \cdot M$ downstream. The total per-round complexity is therefore $O(K \cdot E \cdot B \cdot M + N \cdot d^3)$. From the above analysis, it can be concluded that the worst time complexity of FedEMoE is less than FedTGP but greater than other methods.

### D.7 RELATED TECHNIQUE

#### D.7.1 MoE ADJUSTMENT

Table 9: Comparison of dynamic-capacity MoE mechanisms. *Comp.* = additional compute cost per round. *Scene* = centralised vs federated. *Data* = extra public/validation data required. *Feat.* = whether the operation breaks the existing feature space and demands full-model re-warming.

| Method | Comp. | Scene | Data | Feat. |
|---|---|---|---|---|
| BASE Layer | Low | Centralized | Validation set | Yes (whole layer re-init) |
| Switch-Transformer | Medium | Centralized | None | Yes (new FFN weights) |
| ST-MoE | High | Centralized | Gradient buffer | Yes (expert re-randomize) |
| **FedEMoE (ours)** | **Low** | **Federated** | **None** | **No (sub-expert grafting)** |

The fundamental difference lies in feature space continuity. Existing dynamic MoE methods (BASE (Lewis et al., 2021), Switch-Transformer (Fedus et al., 2022), ST-MoE (Zoph et al., 2022)) rely on destructive capacity adjustments—re-initializing, replacing, or re-randomizing weights. This effectively 'resets' the expert's learning, necessitating heavy re-training, large gradient buffers, or global warm-up cycles to recover.

In contrast, FedEMoE employs preservative elasticity. By performing lightweight SVD on the SE weight matrix locally, we decompose and restructure experts without destroying the learned feature manifold. This allows us to adjust capacity while keeping PEG weights intact, eliminating the need for global re-warming or uploading extra data—a critical requirement for federated learning contexts.

#### D.7.2 DIAGNOSTIC TECHNIQUE

Table 10: Comparison of expert-state diagnostic tools.

| Method | Needs Extra Data | FL-Friendly |
|---|---|---|
| Validation-loss trigger | held-out set | medium |
| Gradient-norm / Fisher | batch gradients | poor |
| Fixed schedule | none | medium |
| **Weight-spectrum (ours)** | **none** | **best** |

Weight-spectrum analysis is the only diagnostic that requires no extra data or labels, uploads no additional tensors, and provides a provable upper bound on client drift. These properties make it the most communication-efficient, privacy-preserving, and theoretically grounded choice for elastic capacity control in federated learning.

# E IMPLEMENTATION DETAILS

## E.1 HYPERPARAMETERS SETTINGS

All experiments were conducted using PyTorch version 2.1.0 on NVIDIA 4090D GPUs with CUDA 12.1. Key hyperparameters included:

- Optimizer: Adam for client updates.

- Client Local Training Learning Rate ($\eta$): 0.005

- Local Epochs: 1

- Batch Size (Local Training): 64

- Data Size of Knowledge Exchange: 64

- KD-Temperature (**T**): 2.0

- Shared expert's MoE Experts ($N$): 3

- Shared expert's MoE top$K$: 2

- Communication Rounds ($T$): 100

- Client Participation Rate: 0.2

- Data Heterogeneity ($\beta$ for Dirichlet): 0.01, 0.1, 0.5, 10

- The gap of scaling operations ($\mathcal{T}$): 20

- The initial expansion threshold ($v_0$): 0.8

- The smoothing factor ($\gamma$) of EMA: 0.07

- The energy ratio ($\theta$): 0.99

- The fraction of top components to drop ($\alpha$): 0.9

- The threshold of tail energy proportion: 0.05

- The random seed: 1

## E.2 MODEL SETTINGS

In our main experiments, the models are based on the MoE architecture. This choice leads to model heterogeneity (Table 11) that comes from several factors. The main ones are the differences in model structure caused by varying the number of experts in the MoE setup (Table 12) and the number of experts activated during processing. Then there's the personalized composition of expert groups, which varies across models. Each model has its own combination of expert types. Also, the internal structures of the experts in Table 13 and classifiers themselves can differ, affecting how they process features and make decisions. In terms of model size, the models we set up in our experiments are similar to ResNet18 (11.2MB). For other backbone networks (ResNet, MobileNet, GoogleNet), we replace the first fully connected layer in their network with the MoE layer in the Table 12.

Table 11: CNN Configurations used in Heterogeneous Experiments.

| Name | Component | Params |
|---|---|---|
| $CNN_1$ | 1 Conv layer, $MoELayer_1$, 1 FC Layer | 10.44M |
| $CNN_2$ | 1 Conv layer, $MoELayer_2$, 1 FC Layer | 11.37M |
| $CNN_3$ | 1 Conv layer, $MoELayer_3$, 1 FC Layer | 14.35M |
| $CNN_4$ | 1 Conv layer, $MoELayer_4$, 1 FC Layer | 12.08M |
| $CNN_5$ | 1 Conv layer, $MoELayer_5$, 1 FC Layer | 7.06M |

Table 12: MoE Configurations used in Heterogeneous CNNs.

| Name | Component | (N,topK) | Params |
|------|-----------|----------|--------|
| $MoELayer_1$ | 3 $Expert_1$, 1 $Expert_2$,1 $SE(e^*)$ ,1 FC Layer | (5,4) | 10.31M |
| $MoELayer_2$ | 2 $Expert_1$, 1 $Expert_2$,1 $Expert_3$, 1 $SE(e^*)$, 1 FC Layer | (5,3) | 11.25M |
| $MoELayer_3$ | 1 $Expert_1$, 1 $Expert_2$,3 $Expert_3$, 1 $SE(e^*)$, 1 FC Layer | (6,4) | 14.21M |
| $MoELayer_4$ | 2 $Expert_2$, 2 $Expert_3$, 1 $SE(e^*)$, 1 FC Layer | (5,3) | 11.95M |
| $MoELayer_5$ | 2 $Expert_2$, 1 $SE(e^*)$, 1 FC Layer | (2,1) | 7.01M |
| $SE(e^*)$ | 3 $Expert_4$, 1 FC Layer | (3,2) | 0.05M |

Table 13: Expert Configurations used in Heterogeneous MoE layer.

| Name | Component | Params |
|------|-----------|--------|
| $Expert_1$ | 2 Conv layers, 1 FC layer | 0.85M |
| $Expert_2$ | 1 Conv layers, 1 FC layer | 1.07M |
| $Expert_3$ | 1 FC layer1 | 2.01M |
| $Expert_4$ | 1 FC layer1 | 0.02M |

# F  ADDITIONAL RESULTS

## F.1  MAIN EXPERIMENTS

Experimental results in Table 14 on CIFAR-10, CIFAR-100, and Tiny-ImageNet—under both heterogeneous and homogeneous model settings with $K = 20$ demonstrate that the proposed method achieves the highest test accuracy and significantly outperforms existing federated learning algorithms. Specifically, on CIFAR-10, it surpasses the runner-up by 3.21% (heterogeneous) and 2.01% (homogeneous); on the more challenging CIFAR-100, the gains reach 21.07% and 24.41%; and on Tiny-ImageNet, it leads by 11.18% and 3.33%, respectively. Collectively, these results underscore the superior generalization capability of our approach.

| Algorithm | CIFAR-10 | CIFAR-100 | Tiny-Imagenet |
|-----------|----------|-----------|---------------|
| **FedAvg** | 78.79 | 38.54 | 22.74 |
| **FedGC** | 80.57 | 40.33 | 25.83 |
| **FedALA** | 79.64 | 35.06 | 15.86 |
| **FedPAC** | 78.62 | 33.42 | 20.61 |
| **FedDBE** | 79.12 | 38.65 | 22.85 |
| **Ours** | 83.16(3.21%) | 48.83(21.07%) | 28.72(11.18%) |
| **FedKD** | 76.81 | 33.62 | 18.24 |
| **FedProto** | 70.74 | 29.12 | 6.93 |
| **FedMRL** | 81.84 | 39.19 | 18.74 |
| **FedTGP** | 68.42 | 35.64 | 12.61 |
| **FedSA** | 76.01 | 39.41 | 28.18 |
| **Ours** | 83.48(2.01%) | 49.03(24.41%) | 29.12(3.33%) |

Table 14: Test accuracy(%) of client with K = 20 under different datasets in model-heterogeneous and model-homogeneous scenarios.

In the setting of $K = 100$ clients, we evaluate the test accuracies of all methods on CIFAR-10, CIFAR-100, and Tiny-ImageNet under both heterogeneous and homogeneous model scenarios. As shown in Table 15, our method attains 78.26% and 78.84% in the heterogeneous and homogeneous cases, respectively, outperforming the runner-up FedGC by 0.38% and 0.74%. On CIFAR-100, it achieves 41.57% and 41.76%, surpassing FedGC by 16.54% and FedTGP by 18.67%. On Tiny-ImageNet, our method reaches 25.35% in the heterogeneous scenario, a 10.84% gain over FedAvg (22.87%), and 25.82% in the homogeneous scenario, an improvement of 48.05% over FedSA (17.43%).

Table 15: Test accuracy(%) of client with K = 100 under different datasets in model-heterogeneous and model-homogeneous scenarios.

| Algorithm | CIFAR-10 | CIFAR-100 | Tiny-Imagenet |
|---|---|---|---|
| FedAvg | 75.42 | 31.36 | 22.87 |
| FedGC | 77.96 | 35.67 | 21.65 |
| FedALA | 76.23 | 31.41 | 20.16 |
| FedPAC | 75.93 | 16.24 | 17.63 |
| FedDBE | 76.07 | 31.16 | 18.81 |
| **Ours** | 78.26(0.38%) | 41.57(16.54%) | 25.35(10.84%) |
| FedKD | 71.57 | 29.41 | 17.44 |
| FedProto | 68.38 | 23.11 | 1.51 |
| FedMRL | 78.26 | 35.52 | 17.21 |
| FedTGP | 68.13 | 35.19 | 11.33 |
| FedSA | 57.48 | 31.65 | 17.43 |
| **Ours** | 78.84(0.74%) | 41.76(18.67%) | 25.82(48.05%) |

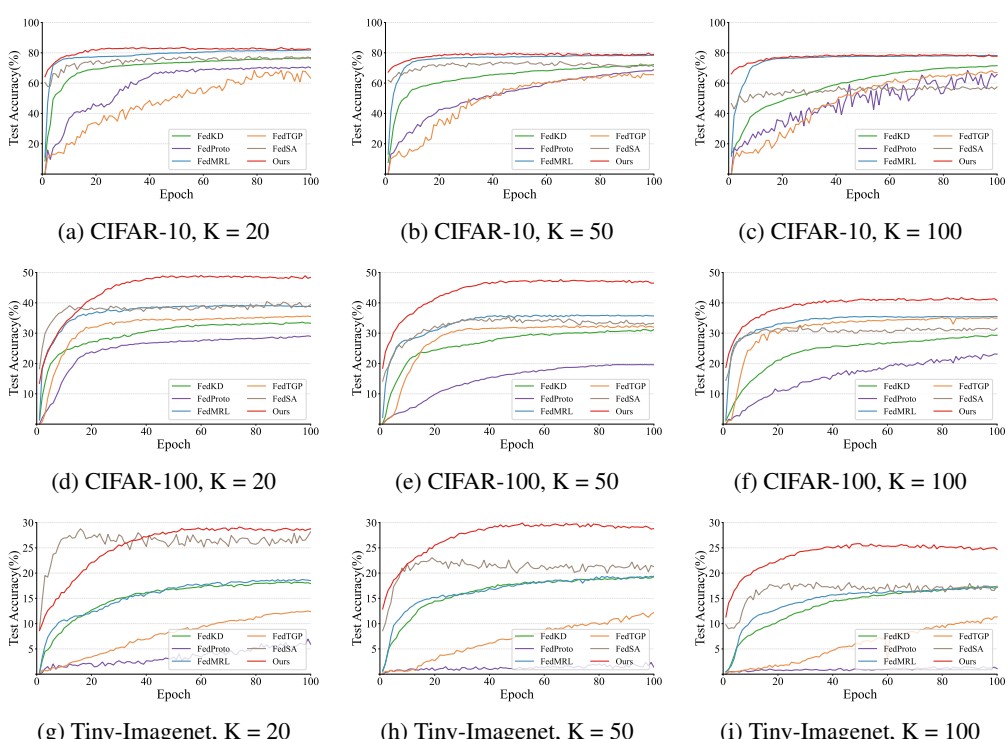

(a) CIFAR-10, K = 20    (b) CIFAR-10, K = 50    (c) CIFAR-10, K = 100

(d) CIFAR-100, K = 20    (e) CIFAR-100, K = 50    (f) CIFAR-100, K = 100

(g) Tiny-Imagenet, K = 20    (h) Tiny-Imagenet, K = 50    (i) Tiny-Imagenet, K = 100

Figure 8: Comparison of Federated Learning Approaches' Performance Across Datasets and Client Scales.

### F.2 CONVERGENCE SPEED ANALYSIS

On CIFAR-10 with 20 clients, the accuracy curves of FedKD, FedTGP, FedProto, FedSA and FedMRL drift around 60% as the communication rounds advance from 0 to 80. In contrast, our approach (Ours) surpasses 80% within only 20 rounds and stays flat thereafter, opening a clear margin of 10–20%. When the client scale grows to 50, all baselines slow down further and still plateau near 70%. However, our approach breaks the 80% barrier around round 20 and finishes more than 15% ahead, demonstrating excellent scalability. In the extremely sparse 100-client setup, the rival curves oscillate violently and FedKD, FedProto, etc. even drop below 60%. Ours remains stable, climbing to 80% around round 20 and maintaining a final gap of roughly 20%, proving its robustness under high decentralization.

Switching to the harder CIFAR-100 task, top-1 accuracies plunge below 50% for 20 clients; FedMRL and FedSA linger around 40%. Ours is the first to exceed 50% within the same schedule and ends up about 10% higher, revealing stronger discrimination for fine-grained classes. At 50 clients on CIFAR-100, the baselines stay in the 30%–40% band while Ours crosses 50% shortly after round 20 and keeps a 15% lead at the end, validating superior generalization when both class count and data heterogeneity increase. Even in the extreme 100-client CIFAR-100 scenario, Ours still reaches above 45% around round 20 whereas FedKD, FedProto, etc. remain near 30%, yielding a terminal advantage of about 15% and highlighting highly efficient knowledge aggregation under severe fragmentation.

Moving to the larger-scale Tiny-ImageNet experiments, none of the baselines exceed 25% with 20 clients; Ours breaks 30% around round 20 and finishes roughly 5% ahead. When clients increase to 50 and 100, the competitors never surpass the 25% ceiling, while Ours stays consistently above 30% and holds a stable 5–7% margin, confirming steady gains on high-resolution, thousand-class data.

Across all tested conditions, Ours consistently delivers faster convergence, higher final accuracy and smoother curves, comprehensively outperforming state-of-the-art alternatives and offering a reliable solution for real-world cross-device deployment.

## F.3 DECOUPLING ABILITY EXPERIMENT

We compare FedEMoE with other decoupling-based PFL methods(DualFed (Zhu et al., 2024), GPFL (Zhang et al., 2023b), FedDecomp (Wu et al., 2024), FedCAC (Wu et al., 2023)) that also separate personal and shared components. Table 16 reports CIFAR-100 accuracies with homogeneous CNNs (K=50, Dir(0.1)). FedEMoE outperforms the strongest competitor FedCAC by 0.54% and beating DualFed by 4.15%, which shows that our spectrum-driven elastic SE extracts richer general knowledge than the fixed-capacity shared branches used in prior methods.

Table 16: Comparison with decoupling-based PFL methods.

| Method | DualFed | GPFL | FedDecomp | FedCAC | **Ours** |
|---|---|---|---|---|---|
| Accuracy (%) | 42.32 | 31.74 | 36.79 | 45.93 | **46.47** |

## F.4 SENSITIVITY ANALYSIS

Table 17 explores the effect of the knowledge-distillation temperature $\mathcal{T}$. We keep the rest of the model fixed at three shared expert $N = 3$ and $topK = 2$, and we vary $\mathcal{T}$ from 1 to 4. The highest accuracy (49.03%) is reached when $\mathcal{T} = 2$. A lower temperature produces an overly sharp probability vector that limits guidance, whereas a higher one flattens the teacher signal and slows convergence. The 0.3% spread across the whole range shows that the method is not sensitive to the exact value.

Table 17: Sensitivity to KD temperature $\mathcal{T}$ (3 experts, topK=2).

| $\mathcal{T}$ | 1 | 2 | 3 | 4 |
|---|---|---|---|---|
| Accuracy (%) | 48.71 | **49.03** | 48.98 | 48.96 |

