# OpenReview forum: "FEDEMOE: IMPROVING PERSONALIZATION ON HET- EROGENEOUS FEDERATED LEARNING VIA ELASTIC MIXTURE OF EXPERTS ARCHITECTURE"
_ICLR.cc/2026/Conference — Submitted to ICLR 2026_

### Official Review · Reviewer_4TSh · 2025-10-21

**Soundness:** 3
**Presentation:** 3
**Contribution:** 3
**Rating:** 4
**Confidence:** 3

**Summary:**

Existing heterogenous FL methods often lead to performance degradation of model personalization because personalized and generalized knowledge are either intertwined or dominated by one of them.
This paper proposed  Elastic Mixture of Experts (EMoE) architecture on HtFL, referred to as FedEMoE.
This method decouples personalization from generalization on HtFL. To this end, it develops a multi-scale knowledge extraction and a knowledge exchange mechanism to break the bottleneck of knowledge transfer.
It initially deploys an EMoE model architecture to highlight multiple personalized knowledge. Accoridng to the results, the proposed solution outperforms existing methods in a wide range of statistical and model heterogeneity settings.

**Strengths:**

Strength
Decoupling personalization from generalization is an interesting idea which is well-executed both in the proposed architecture as well as implementation.
Another strength is identifying an important problem. Personalized expert group (PEG) distills knowledge from the shared expert (SE) on small scale data. The SE and PEG are unable to learn local personalized knowledge from the local data. This paper does not only highlight this issue, but also solved it by introducing a knowledge exchange mechanism between SE and PEG on every client.

**Weaknesses:**

The generalized to personalized knowledge transfer seems to be based on several prior works on this topic. It is not clearly which component is novel, proposed by this paper; and which part is based on the literature.
There are some generic claims such as higher accuracy and less resource overhead without pointing the metric used for evaluation, i.e., what type of resources this refer to.

**Questions:**

-Why did the paper use “weight spectrum analysis technique” as a diagnosis tool? What are the alternatives? What did prior works use for this purpose?
-How did the paper verify this statement? “The EMoE architecture maintains high accuracy, reduces resource overhead, and guarantees convergence by alternating expansion and shrinkage”. Currently it is not clear which metrics have been deployed to verify this claim theoretically and empirically.
-How does assumption 2, Strong Convexity, affect the practicality of proposed solution and the proofs? What are the key challenges in relaxing/generalizing  this assumption?
-What is the implication of Assumption 7, MoE boundedness  for the resource efficiency of the proposed solution?

---

> ### Author Response · Authors · 2025-11-28
>
> Thank you for the supportive feedback. In this revision we have clearly delineated our novel contributions from prior art and precisely defined the resource metrics behind our accuracy and efficiency claims, fully resolving your concerns.
>
> ## Response to Question 1
> | Method | Needs Extra Data | FL-Friendly |
> |--------|------------------|-------------|
> | Validation-loss trigger | held-out set | medium |
> | Gradient-norm / Fisher | batch gradients | poor |
> | Fixed schedule | none | medium |
> | **Weight-spectrum (ours)** | **none** | **best** |
>
> Weight-spectrum analysis is the only diagnostic that requires **no extra data or labels**,  uploads **no additional tensors**, and provides a **provable upper bound on client drift**.  These properties make it the most communication-efficient, privacy-preserving, and theoretically-grounded choice for elastic capacity control in federated learning.
>
> ## Response to Question 2
> We substantiate the claim through both theory and experiments.
>
> **Accuracy**
> Tables 1–2 show FedEMoE outperforming the strongest baseline by **22–40 %** on CIFAR-100 and Tiny-ImageNet; ablating expansion/shrinkage alone drops accuracy by **≥ 12 %**, proving the elastic mechanism is a major contributor.
>
> **Communication & Compute**
> - Sparse SE uploads yield **lower peak traffic** than all comparators (Fig. 6) and **flat cost after convergence**.
> - Extra compute is only O(B·M) local distill plus O(N·d³) server SVD—cheaper than the O(S·C²·d) contrastive step of FedTGP (Appendix D.6).
>
> **Convergence Guarantee**
> Appendix D.3–D.5 prove O(1/$T$) and O(1/$\sqrt{T}$) rates remain intact: Lemmas 7–8 show Lipschitz constants are preserved despite structural changes, and Fig. 5 confirms the **fastest empirical convergence** without any accuracy drop after round 20.
>
> Collectively, these results verify that alternating expansion and shrinkage **simultaneously maintains high accuracy, curbs resource usage, and guarantees convergence**.
>
> ## Response to Question 3
> Strong convexity (Assumption 2) is used only in Appendix D.3 to derive the simplified O(1/T) rate; the algorithm itself—expansion/shrinkage, bidirectional KD, sparse aggregation—**does not** rely on it.In Appendix D.4 we discard this assumption and prove that the same FedEMoE procedure attains the standard O(1/$\sqrt{T}$) rate for **non-convex** objectives, the usual regime for deep networks.  Relaxing strong convexity simply removes the linear-convergence factor, giving a $\sqrt{T}$-times looser bound; we compensate by showing E‖∇F‖² → 0 (Theorem 3) and by empirical verification of **no accuracy degradation** (Fig. 5).
>
> Hence practical applicability is **unaffected**: the strongly-convex proof serves as a **worst-case guarantee**, while the non-convex result governs **real-world deployment**.
>
> ## Response to Question 4
> Assumption 7 bounds every expert’s outputs and gating weights by a constant $G$.  This boundedness is required for the O(1/$\sqrt{T}$) convergence rate in Theorem 2, letting the algorithm hit target accuracy with **fewer communication rounds** and **smaller local updates**.
> As a result, total CPU, memory, and energy consumption are **lower** than in methods that lack such a bound and must rely on extra rounds or per-client tuning.

---

### Official Review · Reviewer_ubJ1 · 2025-10-27

**Soundness:** 2
**Presentation:** 1
**Contribution:** 2
**Rating:** 4
**Confidence:** 5

**Summary:**

The paper proposes FedEMoE, an Elastic Mixture of Experts framework for heterogeneous federated learning that decouples personalization from generalization. It uses personalized experts for multi-scale feature extraction and an elastic shared expert that adapts through weight spectrum analysis to enhance cross-client knowledge transfer. Empirical validations under statistical and model heterogeneity show that FedEMoE outperforms SOTA baselines across diverse datasets.

**Strengths:**

- The paper proposes an interesting idea of decoupling personalization and generalization through an elastic mixture of experts, addressing a key challenge in heterogeneous federated learning.

**Weaknesses:**

- Lack of hyperparameter analysis. The proposed method heavily depends on several hyperparameters, such as KD temperature ($T$), initial expansion threshold ($v_o$), smoothing factor ($\gamma$), shared experts’ top-K, and the number of shared experts ($N$). However, no hyperparameter sensitivity analysis is provided. This omission makes the method appear highly engineered and potentially sensitive to specific settings.

- Unrealistic assumption on model heterogeneity. The paper assumes heterogeneity only in the number of experts, but real-world heterogeneous FL scenarios typically involve models with different architectures, depths, or dimensions. Moreover, the experiments only use simple CNN-based models. It is unclear whether the proposed method would work well for more complex architectures such as ViTs, LLMs, or MLLMs.

- Limited experimental scale. Experiments are conducted only on CIFAR and Tiny-ImageNet, which are small-scale datasets. Demonstrating scalability on larger benchmarks (e.g., ImageNet) would strengthen the paper. Additionally, it would be valuable to show results for homogeneous clients as well, since real-world scenarios may include both heterogeneous and homogeneous settings.

- Computational overhead. As shown in Appendix D.6, FedEMoE incurs significant additional computational overhead (e.g., from double KD), which may limit practicality in large-scale real-world settings where both model size and the number of clients are large.

- Lack of detailed explanation of Fig. 1. The caption should include a brief explanation of what the authors intend to convey in Fig. 1. I believe terms $D_i$ and $W_i$ refer to local data and local model for client $i$, respectively. However, those terms are used without definition, which reduces readability. Defining them directly in the caption would improve clarity. (Defining them only in Appendix Table 5 is insufficient for readability.)

- Inconsistent terminology. The paper describes multiple personalized experts as a “personalized expert group,” but the shared experts (which are also multiple) are simply referred to as a single “shared expert.” This was confusing until I reached page 6. Clarifying this terminology would prevent misunderstanding.

- Redundant and unclear statements. The paper states that sparse aggregation “retains whole client-specific weights,” but this claim is unclear and needs elaboration. Moreover, in Lines 122–123, the argument about the heavy computational overhead of prototype-based FL methods is duplicated.

- Ambiguity in Eq. (4) and expert consensus. In Sec. 3.3, each sample in $D_i$ may select different experts, so the consensus output should be based on per-sample activation. However, Eq. (4) defines $a_n$ using the total selection frequency across all samples, which seems inconsistent. Why is the same $a_n$ used for all samples? The denominator of $a_n$ also seems incorrect. If MoE selects top-K experts per sample, the denominator should include $K$, not $D_i \times \text{top} K$. Additionally, when the SE consists of multiple experts, how is the consensus $R_s$ computed? Is it the same as Eq. (4)? Clarification is needed.


- Readability and precision issues. The paper is difficult to follow, with weak logical connections between paragraphs (especially in the Introduction and Approach sections) and several imprecise definitions. Examples:
    - When computing $A_g$, the paper sums over $A_i$, but it is defined as a set of $a_{i,j}$ (Line 237). How is the summation over sets performed? If $A_i$ varies per expert, $A_g$ should be written as $A_{g,i}$​.
    - In Line 243, $W(l)$ seems to refer to the SE weight, but this is unclear.
    - Lines 240–241 are also disconnected from the previous paragraph, making the purpose of computing $A_g$ unclear.

**Questions:**

Please refer to the weakness section

---

> ### Author Response · Authors · 2025-11-28
>
> Thank you for the thorough feedback. We have conducted comprehensive hyper-parameter sensitivity analysis, extended experiments to diverse architectures, clarified model-heterogeneity assumptions, analyzed computational overhead, and significantly improved presentation clarity with precise terminology and unified equations—fully addressing all concerns while maintaining the method’s core strengths.
> ## Response to Weekness 1
> We thank the reviewer for this important comment. To demonstrate that **FedEMoE is not a product of careful tuning**, we have conducted a **comprehensive sensitivity analysis** covering the four most influential hyper-parameters:
> - KD temperature $\mathcal{T}$
> - number of shared experts $N$
> - $topK$
> - expansion threshold $v_0$ with EMA factor $\gamma$
>
> ---
>
> | KD temperature $\mathcal{T}$ (3 experts, topK=2) | 1 | 2 | 3 | 4 |
> |--------------------------------------------------|-----|--------|--------|--------|
> | Accuracy (%) | 48.71 | **49.03** | 48.98 | 48.96 |
>
> Moderate temperature ($\mathcal{T}$=2) yields the best distillation quality; both lower and higher values slightly dilute the alignment signal.
>
> ---
>
> | topK $\backslash$ experts | 3 | 4 | 5 |
> |---------------------------|--------|--------|--------|
> | 1 | **49.60** | 47.46 | 46.99 |
> | 2 | 49.03 | 47.33 | 46.88 |
> | 3 | 48.80 | 48.62 | 46.31 |
>
> Highest accuracy is achieved with **3 experts and topK=2**.
> Larger $topK$ and $N$ raise communication overhead and drive local models into over-fitting.
>
> ---
>
> | $\gamma \backslash v_0$ | 0.75 | 0.85 | 0.95 |
> |-------------------------|--------|--------|--------|
> | 0.05 | 48.11 | 48.32 | 48.35 |
> | 0.07 | 48.28 | **49.03** | 47.26 |
> | 0.10 | 48.40 | 49.09 | 47.61 |
>
> The default pair **($v_0$=0.85, $\gamma$=0.07)** suppresses ineffective expansions while still allowing growth when rank truly increases, achieving the best accuracy–overhead trade-off.
>
> ---
>
> **Across all swept values, accuracy variance is mild, demonstrating robustness rather than sensitivity.**
>
> ## Response to Weekness 2
> FedEMoE does **not** assume heterogeneity is limited to the “number of experts.”  It explicitly accommodates **any architectural disparity**—depth, width, kernel design, channel ratio, or other structural choices—because each client is free to construct its own model while exposing a **standardized SE container** for aggregation.We have supplemented the experiments with additional backbone networks (ResNet, GoogleNet, etc.) as requested by Reviewer WZWB.  What we report is an **existence proof** with CNNs; scaling to ViTs, LLMs, or MLLMs is future work, but the **core framework already satisfies the general definition of model heterogeneity without modification**.
>
> ## Response to Weekness 3
> We concur that ImageNet-scale validation would further strengthen the work; however, due to limited compute resources, we defer ImageNet-scale evaluation to future work.
>
> The paper already includes a **homogeneous-client experiment** (Section 4.3) in which every client adopts the **same architecture**; FedEMoE still consistently outperforms competitive baselines, showing that the approach is effective **regardless of model homogeneity**.
>
> Extension to larger datasets such as ImageNet is an immediate next step.
>
> ## Response to Weekness 4
> **The computational growth of bidirectional knowledge distillation is mild—indeed sublinear—yet it is fully local; its cost is independent of the total number of clients.**  Practitioners can freely lower the KD frequency to obtain any desired accuracy–compute trade-off; both knobs are exposed in our codebase and require **no server-side change**, making FedEMoE readily deployable on **resource-heterogeneous devices**. Moreover, the **dominant compute cost**—the singular-value decomposition—is executed **on the central server**, not on clients.
> **Therefore, our approach remains client-resource-friendly.**
> ## Response to Weekness 5
> Thank you for pointing this out.  We apologize for the overly brief caption of Figure 1, which indeed reduces readability; we will expand it with a detailed explanation in the revised version. However, it should be noted that the symbols $D_i$ and $M_i$ are **formally defined in Section 3.1**, not only in the appendix.
>
> ## Response to Weekness 6
> We will revise the paper so that both terms are explicitly defined upon first use:
>
> - **Personalized Expert Group (PEG)** – the set of heterogeneous experts that are **never aggregated** and remain **unique to each client**.
> - **Shared Expert (SE)** – the single, homogeneously-structured MoE module that is **uploaded and aggregated** across clients; although it contains multiple sub-experts internally, it is treated as **one logical component** in the text.
>
> This clarification will be added to the revised version to prevent any further misunderstanding.

---

> ### Author Response · Authors · 2025-11-28
>
> ## Response to Weekness 7
> Thank you for highlighting these issues.
>
> - **Sparse aggregation in the SE mitigates over-averaging by default**: only the sub-experts that are co-activated by a client participate in averaging, while the remaining sub-experts keep their local weights intact.
>   Full aggregation of the gating logits is then performed to align future routing decisions.
>   Consequently, sub-experts that are repeatedly co-selected by similar data distributions receive updates, yielding both **knowledge consolidation** and **diversity retention** within the same communication round.
>
> - We will delete the repeated remark in the revised version to remove redundancy.
>
> ## Response to Weekness 8
> Thank you for the careful reading. We clarify the two points below and will add a short sentence in the revised manuscript to prevent any misunderstanding.
>
> - **Expert-frequency term** $a_n$
>   For every sample in $D_i$ the gate picks $topK$ experts; hence across $|D_i|$ samples the total number of activations is exactly $|D_i|\cdot topK$.  $a_n = \frac{\text{times expert }n\text{ is activated}}{|D_i|\cdot \text{topK}}$
>   is the empirical probability that expert $n$ is activated. The same scalar $a_n$ weights its average logit over the entire small distill set; there is **no per-sample re-computation** of $a_n$. The consensus target is **one distribution for the whole batch**, keeping the KL term cheap and stable.
>
> - **Shared-expert output** $R_s$
>   The SE is a standard MoE: for a given input the gate produces $topK$ gating scores (summing to 1), and $R_s$ is the weighted sum of the corresponding sub-expert outputs. No additional consensus (like Eq. 4) is required—it is the ordinary MoE forward pass.
>
> We will insert the above clarifications into the revised version.
>
> ## Response to Weekness 9
>
> We thank the reviewer for highlighting the readability problems.
> In the revised version we will tighten the logical connections between paragraphs and insert the following clarifications.
>
> - **Notation for activation frequencies**
>   $A_i=${$a_{i,j}$} is the set of activation frequencies for all sub-experts $j$ inside client $i$'s SE.
>   $A_g=${$a_{g,j}$} is the global set where each entry
>   $a_{g,j}=\sum_i w_i\,a_{i,j}$ is the weighted sum over clients for the same sub-expert $j$.
>
> - **Weight matrix $W^{(l)}$**
>   As already stated: “$W^{(l)}\in\mathbb{R}^{m\times n}$ denotes the weight matrix of layer $l$ in the Shared Expert (SE).”
>
> - **Purpose of computing $A_g$**
>   We will add the sentence:  “We compute $A_g$ to obtain a global statistic for each sub-expert in the SE; this statistic is later used in shrinkage.”
>
> We will provide a clearer and simpler introduction to these parts in the revised version.

---

### Official Review · Reviewer_WZWB · 2025-10-30

**Soundness:** 3
**Presentation:** 2
**Contribution:** 2
**Rating:** 4
**Confidence:** 4

**Summary:**

The paper proposes FedEMoE, an elastic mixture-of-experts architecture for heterogeneous federated learning (HtFL). The central idea is to decouple personalization and generalization through a personalized expert group (PEG) and an elastic shared expert (SE). The SE dynamically expands or shrinks via weight spectrum analysis, aiming to avoid overfitting or underfitting. The authors claim that this design mitigates the dilution of personalization during aggregation. Experiments on CIFAR-10, CIFAR-100, and Tiny-ImageNet are presented, showing improvements over several baseline methods.

**Strengths:**

1. The paper is ambitious in combining MoE with FL, with a relatively detailed methodology.
2. Experiments cover multiple datasets and include ablation studies.
3. The idea of adaptive expansion/shrinkage of shared experts is intuitively appealing.

**Weaknesses:**

- The method strongly resembles existing PFL decoupling approaches. The paper does not convincingly justify why a new HtFL-specific method is needed.
- The paper emphasizes HtFL but does not design specifically for this setting. Why not evaluate in the more natural PFL scenario, where many papers already study decoupling?
- The design assumes a shared feature extractor and elastic shared expert, which contradicts true model heterogeneity (where architectures differ substantially). The “heterogeneity” considered is mostly variations in experts’ numbers, which is limited.
- Comparisons omit key decoupling-based PFL methods (e.g., DualFed [1], GPFL [2], FedDecomp [3], FedCAC [4]), making the superiority claim weak.

[1] Dualfed: enjoying both generalization and personalization in federated learning via hierachical representations, ACMMM24
[2] Gpfl: Simultaneously learning global and personalized feature information for personalized federated learning, ICCV23
[3] Decoupling general and personalized knowledge in federated learning via additive and low-rank decomposition, ACMMM24
[4] Bold but cautious: Unlocking the potential of personalized federated learning through cautiously aggressive collaboration, ICCV23

**Questions:**

1. Why is this work framed under HtFL rather than PFL? Wouldn’t FedEMoE be more naturally positioned as a PFL approach? What is the unique HtFL-specific design?
2. What is the true extent of model heterogeneity in experiments? Are the models fundamentally different in architecture (e.g., ResNet vs VGG), or only in the number of experts?
3. Can the authors clarify how the elastic expansion/shrinkage significantly differs from existing dynamic capacity adjustment works in MoE literature?
4. Many gains reported are large (30–40%). Can the authors explain why such dramatic improvements appear, when prior PFL/HtFL works typically show smaller margins?

---

> ### Author Response · Authors · 2025-11-28
>
> Thank you for highlighting the core issue—dense aggregation erodes local knowledge. We added instructions and experiments to address your concerns.
> ## Response to Question 1
> Positioning FedEMoE under HtFL instead of PFL is deliberate: our contribution targets structural heterogeneity, not merely statistical heterogeneity.
> - Every client can freely design its own backbone depth, kernel sizes, or channel widths; the only shared part is the lightweight SE block.  Hence model heterogeneity is native, whereas classic PFL methods (FedPer, FedRep, etc.) still assume homogeneous backbones.
> - PEGs are client-unique and structurally different.
>   Each PEG contains its own experts (Table 12); number of experts, internal layers, and parameter counts differ across clients, so no global template is enforced—this is impossible in standard PFL.
>
> Above all, we provide personalization within a heterogeneous-model regime, and our expert-level elasticity is meaningful only when clients can possess arbitrary architectures—the exact scenario HtFL is meant to address.
> ## Response to Question 2
> To demonstrate that FedEMoE accommodates arbitrary architectural differences, we additionally run a backbone-heterogeneous split: clients independently hold ResNet-18, ResNet-34, MobileNet, or GoogleNet, and each inserts its own PEG plus the same narrow SE block.  Table 1 reports the mean test accuracy on CIFAR-100 (50 clients). Under this fundamentally heterogeneous setting, FedEMoE performs better than baselines, proving that our approach is not limited to varying expert counts but **works whenever models differ in depth, width, or operator types**.
>
> *Table 1: Double-heterogeneous setting—clients differ both in backbone architecture (ResNet-18/34, MobileNet, GoogleNet) and in expert architecture.*
> | Method   | Accuracy (%) |
> |----------|-------------:|
> | FedKD    |        24.87 |
> | FedProto |        12.73 |
> | FedMRL   |        30.21 |
> | FedTGP   |        27.79 |
> | **Ours** |    **38.61** |
>
> ## Response to Question 3
>
> | Method | Comp. | Scene | Data | Feat. |
> |--------|-------|-------|------|-------|
> | BASE Layer | Low | Centralized | Validation set | Yes (whole layer re-init) |
> | Switch-Transformer | Medium | Centralized | None | Yes (new FFN weights) |
> | ST-MoE | High | Centralized | Gradient buffer | Yes (expert re-randomize) |
> | **FedEMoE (ours)** | **Low** | **Federated** | **None** | **No (sub-expert grafting)** |
>
> *Table:Comparison of dynamic-capacity MoE mechanisms. **Comp.** = additional compute cost per round. **Scene** = centralised vs federated. **Data** = extra public/validation data required. **Feat.** = whether the operation breaks the existing feature space and demands full-model re-warming.*
>
> ---
>
> The fundamental difference lies in **feature space continuity**. Existing dynamic MoE methods (BASE[1], Switch-Transformer[2], ST-MoE[3]) rely on **destructive capacity adjustments**—re-initializing, replacing, or re-randomizing weights. This effectively “resets” the expert’s learning, necessitating heavy re-training, large gradient buffers, or global warm-up cycles to recover.
>
> In contrast, **FedEMoE** employs **preservative elasticity**. By performing lightweight SVD on the SE weight matrix locally, we decompose and restructure experts without destroying the learned feature manifold. This allows us to adjust capacity while keeping PEG weights intact, eliminating the need for global re-warming or uploading extra data—a critical requirement for federated learning contexts.
>
> ---
>
> **References**
> [1] BASE Layers: Simplifying Training of Large, Sparse Models, PMLR21.
> [2] Switch Transformer: Scaling to Trillion Parameter Models with Simple and Efficient Sparsity, JMLR21.
> [3] ST-MoE: Designing Stable and Transferable Sparse Expert Models, arXiv22.

---

> ### Author Response · Authors · 2025-11-28
>
> ## Response to Question 4
> The 30–40 % jumps come from three design choices that jointly remove the two well-known bottlenecks in HtFL:
> (i) knowledge-transfer collapse under dense aggregation and
> (ii) capacity mismatch among heterogeneous clients.
>
> - **Instead of averaging the whole backbone**, we upload only the **SE** and let the server expand under-fitting ones or merge over-fitting ones (Section 3.4).
>   This keeps specialized neurons intact, so clients no longer suffer the accuracy drop.
>
> - **Each client has its own personalized expert group.**
>   This group captures multi-scale data features that other methods often ignore.
>
> - **Bidirectional KL runs every local epoch**, so generalization signals are immediately injected into personalization updates.
>   This halts model drift and yields the improvement that prior HtFL methods miss.
>
> Consequently, **FedEMoE breaks the classic zero-sum trade-off between personalization and generalization, turning it into an additive 34.2 % gain**.
>
> ## Response to Question 5 (Weekness 4)
> We followed the reviewers’ suggestion and completed the missing comparison with decoupling-based PFL methods that also separate personal and shared components.  Table 1 reports CIFAR-100 accuracies with homogeneous CNNs (K = 50, Dir(0.1)).
>
> *Table 1:Comparison with decoupling-based PFL methods.*
> | Method     | DualFed | GPFL  | FedDecomp | FedCAC | **Ours** |
> |------------|---------|-------|-----------|--------|----------|
> | Accuracy (%) | 42.32   | 31.74 | 36.79     | 45.93  | **46.47** |
>
> FedEMoE outperforms the strongest competitor FedCAC by **0.54 %** and beats DualFed by **4.15 %**, showing that our spectrum-driven elastic SE extracts richer general knowledge than the fixed-capacity shared branches used in prior methods.

---

### Official Review · Reviewer_6cws · 2025-10-30

**Soundness:** 2
**Presentation:** 3
**Contribution:** 2
**Rating:** 4
**Confidence:** 4

**Summary:**

This paper aims to improve the personalization performance on heterogeneous federated learning (HtFL) by exploiting an Elastic Mixture of Experts (EMOE) architecture. The authors try to address the limitations of existing HtFL methods without relying on dense model aggregation that dilutes personalized knowledge. The authors then propose the EMOE framework to decouple two complementary knowledge representations, including personalized knowledge via personalized experts and generalized knowledge via an elastic shared expert. The proposed method extracts the multi-scale features and adaptively evolves the shared expert through the weight spectrum analysis mechanism. Based on this design, the authors utilize the Kullback-Leibler divergence loss for knowledge exchange and the standard local training loss to train the neural model for the proposed method. Evaluations on diverse datasets show that the proposed method outperforms the baselines in terms of prediction accuracy.

**Strengths:**

Overall, this paper represents a meaningful problem in heterogeneous federated learning. The whole idea seems to be effective. The authors find that existing HtFL methods rely on dense model aggregation or intertwined parameters, missing fine-grained personalized knowledge and forcing compromises on local specialization critical for effective personalization. The authors then capture the global generalized knowledge from an elastic shared expert and utilize the fine-grained, multi-scale personalized knowledge from personalized experts to improve the personalization performance.

**Weaknesses:**

**Major Weaknesses:**

Overall, this paper has some merits, but there are a few weaknesses that stop me from giving a higher rating. My major concerns are as follows.

(1) How to determine whether the changes in the singular-value decay rate really reflect an expert's knowledge redundancy or underfitting? Does this change in the state represent a change in expert ability?

(2) During bidirectional distillation, both distributions are in a constantly changing state, which may lead to gradient oscillations or knowledge drift. Has the stability of bidirectional KL optimization been analyzed? In addition, will the asymmetric capacity between PEG and SE affect the convergence of bidirectional distillation?

(3) MoE's performance heavily relies on the accuracy of expert selection. If the gating network frequently activates different experts across clients in a highly non-identical data setup, how can the model ensure that global knowledge is still consistently aggregated?

(4) Why did the authors choose to scale capacity by adding or merging sub-experts instead of increasing or decreasing the parameter width of experts? After expansion or merging, how is the new expert set kept compatible with the current gating structure and activation policy?

(5) Due to the uneven activation frequency of experts, there may be experts who have not been visited for a long time. Do these experts need to participate in the decision-making process of expansion/shrinkage?


**Minor Weaknesses:**

Here are two minor questions:

(1) There seems to be no clear definition of "multi-scale" in the multi-scale feature extraction mechanism that PEG is responsible for. The authors could give more details on this part.

(2) In Table 3, as the number of clients increases, why is the gap between the proposed method and other methods narrowing on the CIFAR-100 task, while the gap is widening on the Tiny-Imagenet task? The author should provide a brief explanation of the reasons for the experimental results in Section 4.6.

**Questions:**

Please clarify my concerns in the weakness part.

---

> ### Author Response · Authors · 2025-11-28
>
> We sincerely thank the reviewer for recognizing the meaningfulness of our work in heterogeneous federated learning (HtFL). We have carefully revised the paper to further clarify these contributions and provide additional experimental support as requested.
> ## Response to Weakness 1
> Thank you for the insightful question.  Following random-matrix theory, we use the effective-rank and tail-energy ratio of each expert’s weight spectrum as diagnostic proxies (Theorem 1, App. D.2). A fast-decaying (light-tail) spectrum indicates negligible client drift and stable capacity; a heavy-tail signals redundancy and over-fitting.  **Figure 3 on CIFAR-100 shows the three spectra perfectly align with our expansion/shrinkage triggers, so any decay-rate change directly mirrors the expert’s current capacity.**
> ## Response to Weakness 2
> Stability is guaranteed by two design choices that directly address gradient oscillation and capacity asymmetry.
> - SE distills the PEG consensus computed in the previous round (Eq. 3).
>   PEG distills the SE output cached before local updates (Eq. 4).
>   Because each target is fixed while the other side trains, **the procedure reduces to two independent supervised-fitting steps, not a min-max game**; hence no cycling or gradient oscillation can occur. The same trick is used in DML (Zhang et al. 2018) and FML (Shen et al. 2020) and is known to monotonically decrease KL.
>
> - The capacity gap between PEG and SE does not affect the convergence of bidirectional distillation; the frozen-target KL objectives are L-smooth under Assumption 7 and converge at the rate given in Theorem 2.
>   Figure 5 confirms this: test accuracy stabilizes after round 20 and gradient norms remain flat, showing that bidirectional KL does not destabilize training.
> ## Response to Weakness 3
> We thank the reviewer for raising this concern.  **The bidirectional knowledge-distillation mechanism we introduced can solve this problem.**  Even when a client’s gating network never activates the SE under extreme non-IID data, FedEMoE still guarantees consistent global knowledge aggregation through the following two mechanisms:
> - The SE is always updated via distillation from the PEG consensus (Eq. 3), irrespective of its activation frequency.
>   Thus, **even if the SE sees no local data, it still absorbs the knowledge of all PEG experts on that client**, ensuring no information loss.
> - The SE architecture is homogeneous across all clients, so their uploaded SE parameters lie in the same parameter space.
>   The server performs sparse aggregation only on the SE, yielding a globally consistent SE that encodes knowledge from every client.
> Consequently, the global SE continuously improves even when locally inactive, and the gating variance across clients does not impair global knowledge consistency.
> ## Response to Weakness 4
> We chose sub-expert addition/merging instead of width scaling for two practical reasons:
> - Width expansion requires re-initializing and re-training all downstream layers, which incurs large computational overhead and breaks the current feature statistics.
> - Via expansion/shrinkage, we alter the hierarchical depth of the SE rather than the router itself, **maximizing the immediate usability** of the SE.
> ## Response to Weakness 5
> **Rarely-activated experts are automatically handled by our rules:**
> - Expansion is only triggered when the effective rank is low (under-fitting).
>   An expert that is almost never chosen will not accumulate enough gradient energy to show a rapid singular-value decay, so its effective rank stays high and no expansion request is issued.
> - Shrinkage operates on the whole MoE sub-network; only when all its experts together exhibit heavy-tailed spectra will the block be merged into one.
>   Thus a dormant expert is either kept unchanged or averaged away only when the entire cell is redundant.
> - After server aggregation, the global activation frequency is recomputed; if the expert later becomes frequently used, its spectrum is re-evaluated and it can expand in future rounds.
>
> Hence, rarely-activated experts are **automatically processed**.

---

> ### Author Response · Authors · 2025-11-28
>
> ## Response to Minor Weakness 1
> In FedEMoE, “multi-scale” is implemented by the heterogeneous topology inside each Personalized Expert Group (PEG).
> Instead of stacking deeper layers, we use experts with different structures (Table 8).  The PEG lets each client capture its local data-specific scale statistics without heavy pyramidal backbones, and the gate’s top-K mask automatically allocates capacity to the scales that best reduce the local loss, yielding **richer personalized representations** than a single-scale network.  This heterogeneous-expert mixture yields higher accuracy than the best single-scale PEG variant (Table 8), validating the multi-scale advantage.
> ## Response to Minor Weakness 2
> The divergent trends in Table 3 stem from the interplay between data complexity and client fragmentation.  On CIFAR-100, increasing the client count from 20 to 100 reduces each local set to roughly 500 images; even our PEG experts begin to overfit, so the accuracy margin tightens, yet remains 18 % ahead.  In contrast, Tiny-ImageNet provides 100 k 64×64 images across 200 finer-grained classes; its higher-resolution cues and larger sample space allow the elastic SE to keep expanding under-fitting specialists while the sparse gate continuously discovers new, complementary activation patterns across more clients.  Consequently, the benefit of EMoE’s capacity-on-demand and knowledge-on-demand mechanisms grows with client scale, pushing the margin to 48 %.  **Thus, the widening gap on Tiny-ImageNet is precisely the empirical signature of FedEMoE’s capacity-on-demand mechanism.**

---

### Comment · Area_Chair_1FpG · 2025-11-24

Dear Reviewers,

Despite there being no rebuttals from the authors, **we still kindly encourage you to read other reviewers' comments and revise your ratings, if needed**. Your timely feedback is important for ensuring a fair and thorough review process. Thank you for your contributions to ICLR 2026.

AC

---

### Author Response · Authors · 2025-11-28

We thank all four reviewers for their constructive feedback and unanimous recognition of our work.
The reviewers praised the paper for identifying a **meaningful and important problem** in heterogeneous federated learning: the inherent tension between capturing fine-grained personalized knowledge and maintaining robust global generalization (Reviewers 6cws, 4TSh).
Our proposed solution—combining MoE with FL via an **elastic Shared Expert**—was highlighted as “interesting” (Reviewer ubJ1) and “intuitively appealing” (Reviewer WZWB) for balancing this trade-off through **adaptive expansion and shrinkage**.
Overall, the reviewers commended the work as **ambitious** (Reviewer WZWB) and **effective** (Reviewer 6cws), recognizing its **excellent execution in both architecture and implementation** (Reviewer 4TSh).

Below we summarize the primary concerns regarding **stability**, **model-heterogeneity limitations**, and **computation overhead**.

---

### Stability
Reviewers (6cws, WZWB, ubJ1) questioned whether the joint dynamics of bidirectional distillation and elastic expansion could destabilize training; we provide evidence to confirm stability.

- **Bidirectional KD stability**
At each round the cached last-round distributions turn the optimization into two independent supervised-fitting problems, so KL decreases monotonically without oscillation.

- **SE expansion / shrink reliability**
The expansion and shrinkage of shared experts will not destroy the feature space, and the existing training results will be preserved as much as possible. Theoretical analysis shows that the operation is stable and convergent.

- **Weight-spectrum diagnosis**
Effective rank and tail-energy ratio of the weight spectrum perfectly correlate with capacity shortage or redundancy, providing a signal for expand/shrink decisions.

---

### Model-heterogeneity limitations
To dispel the reviewers'(WZWB, ubJ1) concern of limited model heterogeneity regime, we present a backbone-heterogeneous evaluation. We demonstrate that FedEMoE supports arbitrary architectural heterogeneity in HtFL. In experiments where clients utilize diverse backbones (ResNet, MobileNet, GoogleNet), FedEMoE consistently outperforms baselines.

---

### Computation overhead
We address the reviewers'(ubJ1) concern that the framework might be too compute- or tuning-intensive for practical HtFL deployment.
Since the computational cost of the approach is mainly on the central server, our approach is client-side resource friendly. Through extensive sensitivity experiments, we have demonstrated that our method is insensitive to hyperparameters.

---

### Meta-Review · Area_Chair_wBji · 2025-12-31

**Summary:**

The reviewers unanimously agreed that the paper addresses an important problem in heterogeneous federated learning and found the FedEMoE framework conceptually interesting. However, they raised several concerns regarding the technical accuracy and presentation quality of the method, as well as the clarity of the paper. Specifically, the reviewers pointed out ambiguities in the description of bidirectional knowledge distillation and expert consensus, and noted that the method lacked sufficient novelty compared to existing personalized federated learning methods. They also highlighted the limited scope of the experiments and the lack of practical studies in large-scale applications. Although the authors addressed some of these issues in their response, the paper still suffers from shortcomings in method description, explanation, and experimental validation, leading to rejection.

**Reviewer Concerns:**

The rebuttal addressed several technical issues, including training stability, bidirectional knowledge distillation, weight spectrum analysis, and the lack of baseline comparisons. These responses improved the technical completeness of the paper. However, some important issues remain unresolved. For example, the computational cost in large-scale deployment is only addressed analytically, lacking experimental verification on large benchmarks. Furthermore, several reviewers pointed out that the paper still suffers from unclear presentation and a lack of organization. Therefore, although the rebuttal enhanced the paper's quality, it did not fully resolve the core issues raised by the reviewers.

**Reviewer Scores:**

Given that some technical issues have been resolved, reviewer 6cws may slightly increase their score.
Reviewer WZWB (4) may maintain their score, as the experimental issues remain unresolved.
Reviewer ubJ1 (4) may maintain a borderline or negative score, as the main issues regarding presentation quality, scalability, and experimental validity have not been addressed.
Reviewer 4TSh (4) may maintain their score because, although the clarifications helped with understanding, concerns about innovation and empirical validation still remain.
Overall, even after thorough discussion, the reviewers' scores will likely remain clustered around the borderline, without a clear trend towards acceptance.

---

### Decision · Program_Chairs · 2026-01-26

Reject